# Macrophage-specific responses to human- and animal-adapted tubercle bacilli reveal pathogen and host factors driving multinucleated cell formation

Christophe J. Queval[1], Antony Fearns[1], Laure Botella[1], Alicia Smyth[2], Laura Schnettger[1], Morgane Mitermite[2], Esen Wooff[3], Bernardo Villarreal-Ramos[3,4], Waldo Garcia-Jimenez[5¤], Tiaan Heunis[6], Matthias Trost[6], Dirk Werling[7], Francisco J. Salguero[5,8], Stephen V. Gordon[2], Maximiliano G. Gutierrez[1]*

**1** The Francis Crick Institute, London, United Kingdom, **2** UCD School of Veterinary Medicine and UCD Conway Institute, University College Dublin, Dublin, Ireland, **3** Animal and Plant Health Agency, Addlestone, United Kingdom, **4** Institute of Biological, Environmental and Rural Sciences (IBERS), Aberystwyth University, Aberystwyth, United Kingdom, **5** Department of Pathology an Infectious Diseases. School of Veterinary Medicine. University of Surrey, Guildford, United Kingdom, **6** Biosciences Institute, Newcastle University, Newcastle, United Kingdom, **7** Department of Pathobiology and Population Sciences, The Royal Veterinary College, University of London, Hatfield Hertfordshire, United Kingdom, **8** National Infection Service, Public Health England (PHE), Porton Down, Salisbury, United Kingdom

¤ Current address: Ingulados Research, Caceres, Spain
* max.g@crick.ac.uk

**Data Availability Statement:** Mass spectrometric raw data has been uploaded to the ProteomExchang. Project accession: PXD017949

## Abstract

The *Mycobacterium tuberculosis* complex (MTBC) is a group of related pathogens that cause tuberculosis (TB) in mammals. MTBC species are distinguished by their ability to sustain in distinct host populations. While *Mycobacterium bovis* (Mbv) sustains transmission cycles in cattle and wild animals and causes zoonotic TB, *M. tuberculosis* (Mtb) affects human populations and seldom causes disease in cattle. The host and pathogen determinants underlying host tropism between MTBC species are still unknown. Macrophages are the main host cell that encounters mycobacteria upon initial infection, and we hypothesised that early interactions between the macrophage and mycobacteria influence species-specific disease outcome. To identify factors that contribute to host tropism, we analysed blood-derived primary human and bovine macrophages (hMφ or bMφ, respectively) infected with Mbv and Mtb. We show that Mbv and Mtb reside in different cellular compartments and differentially replicate in hMφ whereas both Mbv and Mtb efficiently replicate in bMφ. Specifically, we show that out of the four infection combinations, only the infection of bMφ with Mbv promoted the formation of multinucleated giant cells (MNGCs), a hallmark of tuberculous granulomas. Mechanistically, we demonstrate that both MPB70 from Mbv and extracellular vesicles released by Mbv-infected bMφ promote macrophage multinucleation. Importantly, we extended our *in vitro* studies to show that granulomas from Mbv-infected but not Mtb-infected cattle contained higher numbers of MNGCs. Our findings implicate MNGC formation in the contrasting pathology between Mtb and Mbv for the bovine host and identify MPB70 from Mbv and extracellular vesicles from bMφ as mediators of this process.

(Username: reviewer65235@ebi.ac.uk Password: 5yBTVK8y, S2 Table).

**Funding:** This work was supported by the Francis Crick Institute, which receives its core funding from Cancer Research UK (10092 to MGG), the UK Medical Research Council (10092 to MGG), and the Wellcome Trust (10092 to MGG) and by the Biotechnology and Biological Sciences Research Council (BB/N004574/1 to MGG and SVG), Science Foundation Ireland (SFI/15/IA/3154 to SVG) and a Wellcome Trust PhD studentship (109166/Z/15/A to MM), and EU FP7 Research Infrastructures grant agreement No. FP7-228394 (NADIR to SVG). The funders had no role in study design, data collection and analysis, decision to publish, or preparation of the manuscript.

**Competing interests:** The authors have declared that no competing interests exist.

## Author summary

The identification of host and pathogen factors contributing to host-pathogen interaction is crucial to understand the pathogenesis and dissemination of tuberculosis. This is particularly the case in deciphering the mechanistic basis for host-tropism across the MTBC. Here, we show that *in vitro*, *M. bovis* but not *M. tuberculosis* induces multinucleated cell formation in bovine macrophages. We identified host and pathogen mechanistic drivers of multinucleated cell formation: MPB70 as the *M. bovis* factor and bovine macrophage extracellular vesicles. Using a cattle experimental infection model, we confirmed differential multinucleated cell formation *in vivo*. Thus, we have identified host and pathogen factors that contribute to host tropism in human/bovine tuberculosis. Additionally, this work provides an explanation for the long-standing association of multinucleated cells with tuberculosis pathogenesis.

## Introduction

Host tropism relates to the range of host species that a pathogen can sustain within, requiring the pathogen to infect, replicate and transmit in this host. *Mycobacterium tuberculosis* (Mtb) causes tuberculosis (TB) in humans and is a leading cause of morbidity and mortality worldwide from a single infectious agent, with 1.8 million deaths in 2018 [1]. Mtb is an obligate pathogen with distinct tropism for humans, transmitting between individuals *via* aerosols. Infection generally occurs in terminal lung airways, where the bacillus is taken up by alveolar macrophages before disseminating to other organs. Regardless of which tissues are involved, the immune response against the bacillus progressively leads to the formation of granulomas, where the bacilli can either disseminate from or persist [2–4].

On the other hand, bovine TB is caused by *M. bovis* (Mbv) and shows a broader host tropism, infecting and transmitting between a variety of livestock and wildlife populations [5]. Mbv also poses a risk as a zoonotic pathogen, representing a serious threat to human health [1]. In 2005, the WHO declared bovine TB as the most neglected zoonotic disease threatening human health [6–8].

The genomes of Mbv and Mtb are over 99.9% identical [9]. The main genomic differences between these two pathogens encompass 8 regions of difference (RDs) and over 2000 single-nucleotide polymorphisms (SNPs) [9,10]. While this level of genetic difference is relatively small, it results in major phenotypic variation that ultimately defines host tropism. Comparative analysis revealed 77 and 103 genes upregulated at both the RNA and protein level in Mbv AF2122/97 and Mtb H37Rv, relative to each other, during *in vitro* growth [11]. Proteins showing differential abundance included known virulence factors such as EsxA, EsxB as well as the immunogenic MPB70 and MPB83 [11–15].

In both human and bovine TB, macrophages are the first line of defence as well as one of the main cellular reservoirs of tubercle bacilli [16]. In the lung or other organs, the progressive establishment of a TB immune response leads to the recruitment of immune cells including CD4+ T cells, CD8+ T cells, as well as both pro- and anti-inflammatory macrophages organized in structures called granulomas. The presence of multinucleated giant cells (MNGCs; also called Langhans' cells), containing 20 or more nuclei, represents an important feature of granulomatous inflammation [2,17]. MNGCs are formed by fusion of pro-inflammatory macrophages recruited at the inflammation site, and contribute to the global inflammatory response [18,19]. MNGC formation is a multifactorial process involving cytokines such IFN-γ, IL-4, IL-3, TNF-α, IL-2, M-CSF and GM-CSF as well as other fusogenic molecules including

carbohydrates, lipopeptide or secreted cellular factors and receptors including metalloproteinases, tetraspanins, DC-STAMP and E-cadherin [19–27]. However, none of those fusogenic factors is strictly required to induce MNGC and the process is complex. Although macrophage differentiation and activation is crucial for MNGC formation, it is still unclear whether the direct interaction of mycobacteria with macrophages promote MNGC formation, or if it results from the combination of macrophage activation status with the release of specific mycobacterial antigens [28,29].

Here, we compared two archetypal MTBC host-adapted species, Mtb H37Rv and Mbv AF2122/97 [9,10,30] for *in vitro* interactions with GM-CSF monocyte-derived primary human (hMϕ) or bovine (bMϕ) macrophages. We show that Mtb and Mbv reside in different cellular compartments and differentially replicate in hMϕ, while both strains efficiently replicate in bMϕ. We demonstrate that, both *in vitro* and *in vivo*, Mbv but not Mtb specifically promotes MNGC formation in bMϕ, arguing that this host-specific MNGC generation contributes to the contrasting pathogenesis between Mtb and Mbv in cattle. Finally, we show that the secreted Mbv protein MPB70 and bMϕ-derived extracellular vesicles (EVs) are bacterial and host factors specifically implicated in bMϕ-multinucleation. Our results identify functional differences between Mbv and Mtb host-pathogen interaction, disclosing a role for MPB70 and EVs in the process of multinucleation in bovine macrophages, and revealing distinct evolutionary adaptations of MTBC to specific hosts.

## Results

### Mtb and Mbv differentially replicate within human or bovine macrophages

In order to identify host and bacterial factors during species-specific macrophage interactions, we set up an experimental infection model comparing, side by side, blood-derived primary human or bovine macrophages (hMϕ and bMϕ, respectively) infected with either Mtb or Mbv. First, we optimized the infection ratios for both macrophage models by analysing the ability of hMϕ and bMϕ to phagocytose Mtb-RFP (Mtb-expressing Red Fluorescent Protein) or Mbv-RFP. After 2 h, we observed that Mtb-RFP and Mbv-RFP uptake was similar in both hMϕ or bMϕ. However, bacterial uptake was approximately 10 times higher in bMϕ when compared to hMϕ. In order to normalize the infection protocol, we then infected bMϕ with a lower MOI (MOI of 1) than hMϕ (MOI of 10) (S1A Fig). Following infection, Mtb and Mbv replication in hMϕ and bMϕ were quantified microscopically after 72 h of infection (Fig 1A). In hMϕ, Mtb replicated as previously shown [31] whereas Mbv replication was delayed (Fig 1B). In contrast, both Mtb and Mbv replicated at similar rates in bMϕ up to 72 h (Fig 1C). After 72 h of infection, there was cytotoxicity associated with bMϕ but not hMϕ (S1B and S1C Fig). In hMϕ, cytotoxicity was detectable after 5 days of infection with Mtb, while a significant number of hMϕ infected with Mbv was still detectable (S1C Fig). Next, we assessed the cytotoxicity of Mtb in hMϕ using the cell death reporter Green Live/Dead (S1D Fig). After 5 days of infection, Mtb actively replicated in hMϕ and was associated with cell death while Mbv slowly replicated in hMϕ with limited cell death (S1D Fig). We then infected hMϕ at a low MOI (MOI 2) with Mtb-RFP and Mbv-RFP for 8 days and found that Mbv replication significantly increased after 8 days of infection compared to the basal uptake observed 2 h post-infection (S1E Fig). Thus, the progression of Mbv-intracellular replication in hMϕ suggests that, although Mbv replication is restricted compared to Mtb, hMϕ failed to completely eradicate the infection.

### Intracellular localization of Mtb and Mbv differs in human macrophages

Next, we tested whether the different bacterial replication rates were associated with a differential ability of Mtb or Mbv to evade host macrophage restriction. Although both hMϕ and bMϕ

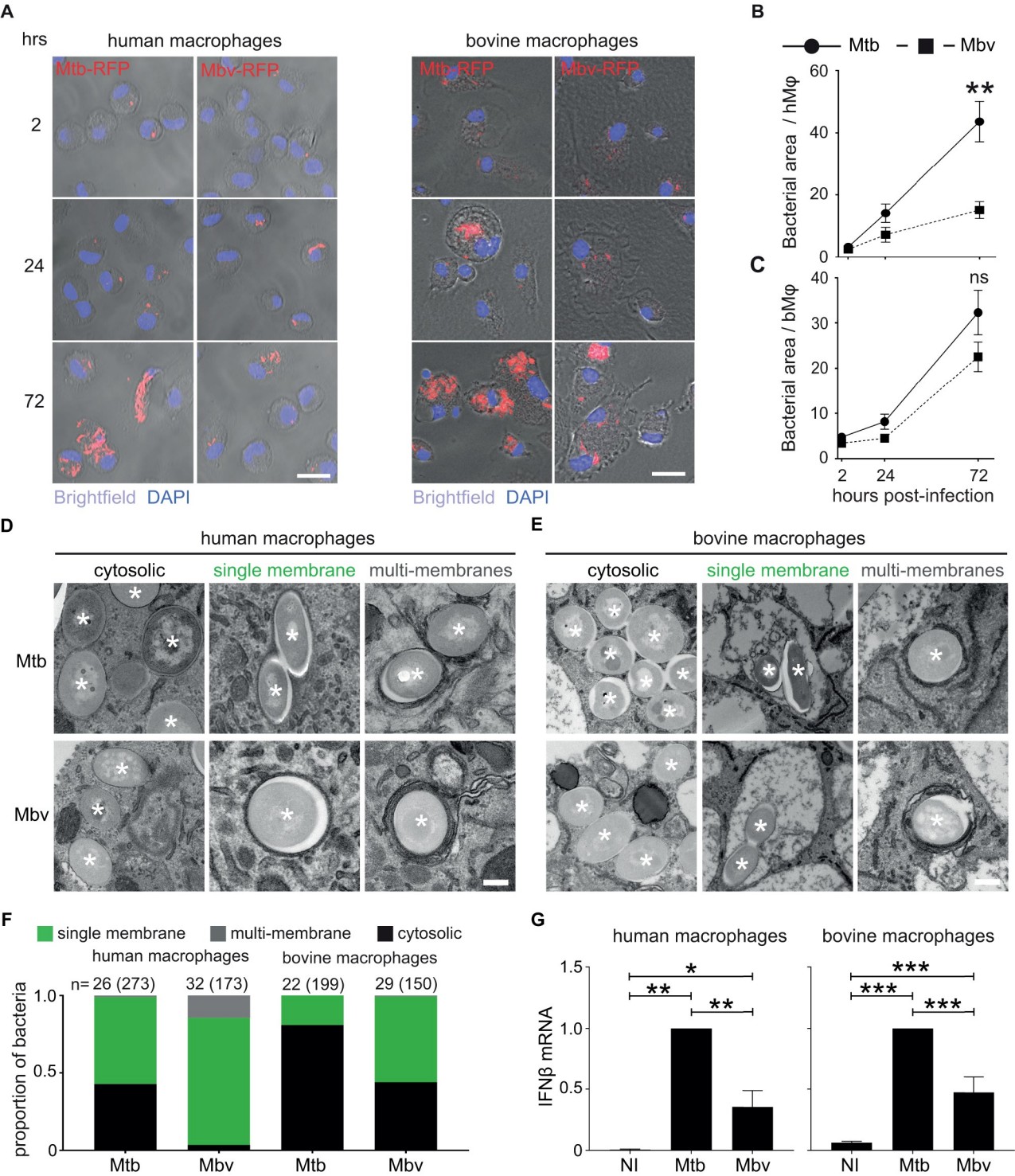

**Fig 1. Mtb and Mbv differentially replicate and localise within human or bovine macrophages.** (A) Confocal images of monocyte-derived GM-CSF differentiated-human or bovine Mφ (hMφ or bMφ) infected with RFP-expressing *M. tuberculosis* H37Rv (Mtb-RFP) or *M. bovis* (Mbv-RFP) for 2, 24 and 72 h. Brightfield was used to visualize the cells. Cell nuclei were stained with DAPI (blue) and bacteria-RFP are visualized in red. Scale bar: 20 μm. (B and C) Quantification of intracellular growth expressed in bacteria area (μm$^2$) per infected cell of Mtb-RFP and Mbv-RFP within hMφ (B) and bMφ (C). (D and E) Electron microscopy images of hMφ (D) and bMφ (E) infected with Mtb or Mbv for 24 h. Asterisks mark the intracellular bacteria. Images were selected to illustrate free cytosolic bacteria (left-hand panels), phagosomal bacteria (middle panels) and bacteria surrounded by multiple membranes (right-hand panels). Scale bar, 200 nm.

(F) Quantification by stereology of the proportion of bacteria contained in each compartment. Black represents the proportion of cytosolic bacteria; green, the proportion of single membrane bound bacteria, and grey, the bacteria surrounded by multiple membranes. "n" represents the number of cells analysed. The total number of bacteria analysed corresponds to the number in brackets. (G) Quantification by RT-qPCR of the relative fold change mRNA expression of interferon-β (*IFNB1*) in hMφ (left graph) and bMφ (right graph) infected with Mtb or Mbv for 24 h. Data are normalized to Mtb. (Housekeeping gene used: *GAPDH*). Non-infected cells (NI) were use as control. For all the figures, p-value is considered significant when < 0.05 and indicate as follow: *p<0.05; ** p<0.01; *** p<0.001; ns: not significant.

were produced with a similar method, electron microscopy analysis showed that the cytoplasm of resting bMφ was strongly enriched with spacious single-membrane vesicles when compared to hMφ (S1F Fig). These large vacuoles were positive for the late endosomal marker, LAMP-1 (S1G Fig). Whereas in hMφ the localisation of intracellular mycobacteria is relatively well characterised [16,32], the localisation of Mbv in bMφ is unknown. We therefore quantified the percentage of bacteria associated with (i) the late endosomal marker LAMP-1, (ii) the pH-sensitive Dye Lysotracker Red (LTR) or (iii) the autophagosome marker LC3B, after 24 h of infection (S2 Fig). For this, we used two different approaches to quantify the localisation: one for markers closely associated with bacteria [33] and another method for spacious compartments such as the LAMP-1 positive phagosomes [34]. In hMφ, 37% ± 14.7 and 39% ± 11.9 of Mtb and Mbv respectively were found associated with LAMP-1. Similarly, in bMφ, 45% ± 9.4 and 36% ± 5.4 of Mtb and Mbv, respectively, were found in a LAMP-1-positive vacuole. The high level of bacterial association with the LAMP-1 marker suggests that, in both species, a significant fraction of the bacteria is membrane-bound (S2A and S2B Fig). In contrast, only 15 to 20% were found associated with lysotracker in both hMφ and bMφ, suggesting that, independent of the host species, a limited proportion of membrane-bound mycobacteria were in acidic compartments after 24 h of infection (S2C and S2D Fig).

In hMφ, LC3B was found associated with 28% ± 6.06 of Mbv, while only 10% ± 0.86 of Mtb was positive for this autophagosome marker (S2E Fig). Autophagy is an anti-mycobacteria pathway [35–37] and the higher association of Mbv with LC3B-positive compartments may explain the delayed intracellular replication. However, Mtb and Mbv presented a similar association with LC3B or all the other markers tested in bMφ (S2F Fig). Next, we analysed the localisation of mycobacteria at the ultrastructural level and defined the proportion of bacteria localised in (i) a single-membrane phagosome, (ii) in a multi-membrane vacuole or (iii) cytosolic (Fig 1D-1F). In hMφ, circa 40% of the Mtb population was localised in the cytosol. Mtb also resided in a single-membrane compartment (>55%) and a minor proportion in multi-membrane compartments (<5%) (Fig 1D and 1F). In contrast, the majority of Mbv were localized in single- and multi-membrane compartments (>75% and around 20%, respectively) and only a minority of Mbv was localised in the cytosol (<5%) (Fig 1D and 1F). On the other hand, in bMφ, the majority of Mtb localised in the cytosol (around 75%) whilst the rest mainly resided in a single-membrane compartment (Fig 1E and 1F). Moreover, Mbv was associated with single-membrane compartments and also localised in the cytosol of bMφ (Fig 1E and 1F). Given that cytosolic DNA will trigger expression of interferon-β (IFNβ) [36,38,39], we monitored by RT-qPCR the expression of *IFNB* in hMφ and bMφ infected with Mtb or Mbv. We found that Mtb induced higher *IFN-β* levels than Mbv in hMφ and bMφ, confirming that, regardless of species, Mtb accessed the cytosol more efficiently than Mbv (Fig 1G).

## Mbv specifically induces the formation of multinucleated giant cells (MNGCs) in bMφ

The similar features observed for both pathogens during infection of bMφ did not reflect the attenuated phenotype of Mtb seen in cattle [40]. However, we noticed that the fusogenic properties of bMφ were different after infection with either Mtb-RFP or Mbv-RFP. As early as 24 h

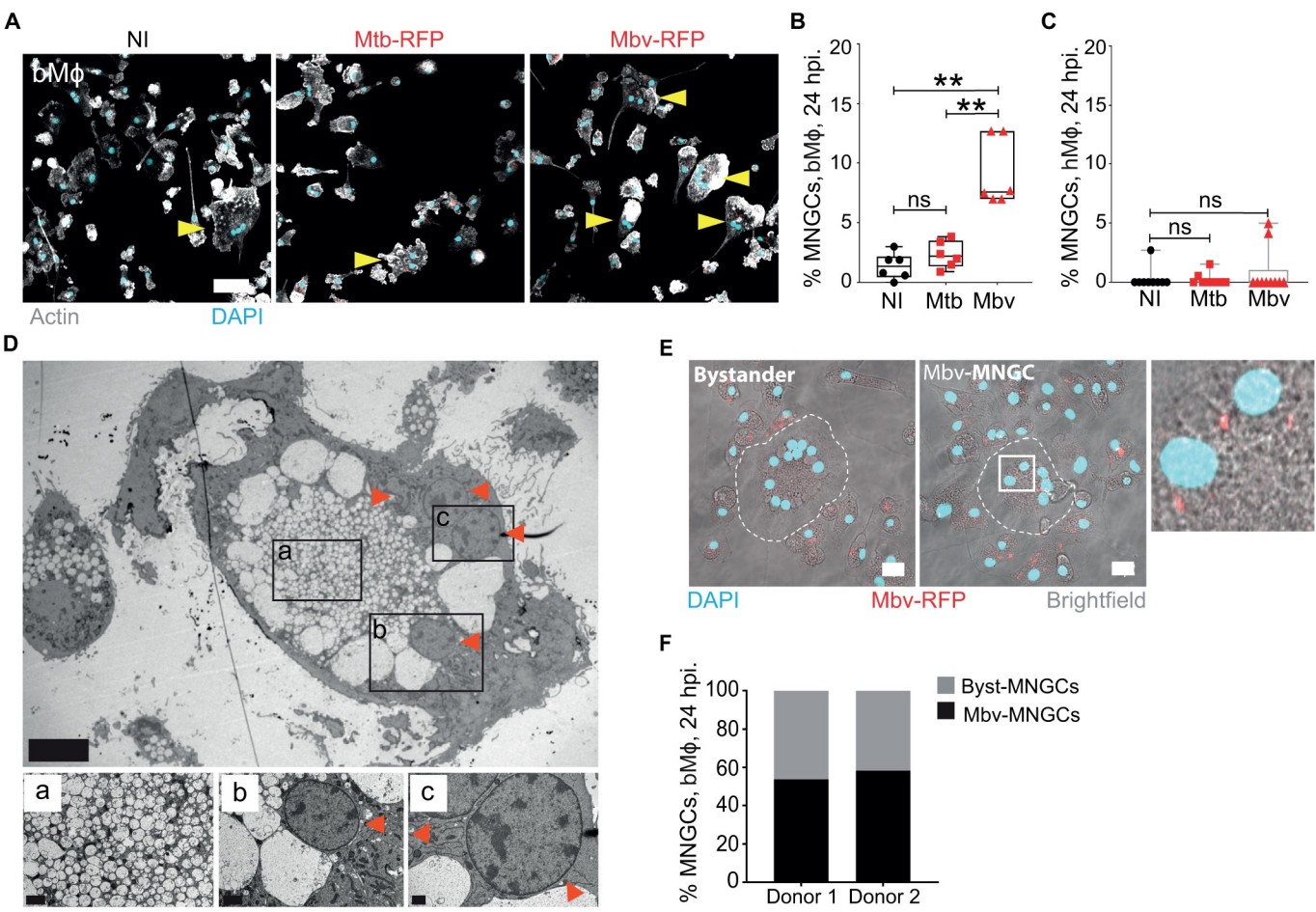

**Fig 2. Mbv specifically induces the formation of MNGCs in bMφ.** (A) Fluorescence confocal images of bMφ infected with Mtb-RFP or Mbv-RFP for 24 h. Non-infected cells (NI) were used as a control. The bacteria are visualized in red, the cell actin cytoskeleton is in white (phalloidin-488) and cell nuclei (DAPI) in cyan. Arrows highlight the cells containing 2 or more nuclei, corresponding to multinucleated giant cells (MNGCs). Scale bar, 50 μm. (B) Quantification of the percentage of MNGCs in bMφ infected with Mtb-RFP or Mbv-RFP for 24 hours. Non-infected cells (NI) were used as a control. Each dot represents one donor tested. (C) Quantification of the percentage of MNGCs in hMφ infected with Mtb-RFP or Mbv-RFP for 24 h. Non-infected cells (NI) were used as a control. (D) Electron microscopy image of an uninfected bystander-MNGC containing four distinct nuclei (marked by red arrows). The images displayed on the right side correspond to the magnification of each region (a-c) delimited by black squares in the main image. Scale bar are D: 40 μm; Dᵃ: 2 μm; Dᵇ: 2 μm; Dᶜ: 1 μm. (E) Maximum projection (14 Z-stacks with an interval of 0.8 μm) from confocal images of bovine MNGCs without intracellular Mbv-RFP (Byst-MNGC) or containing intracellular Mbv-RFP (Mbv-MNGC). In the latter image, the presence of intracellular bacteria is highlighted in the region delimited by a white square, magnified in the right corner. Bacteria are visualized in red, the cells with brightfield image, and nuclei in cyan. A dashed line delimits the edge of MNGCs. (F) Quantification of the percentage of bystander and Mbv-MNGCs in the bMφ population for two independent donors. Each bar chart represents the total MNGC population (100%); in black is percentage of Mbv-infected MNGCs, and in grey the percentage of uninfected bystander-MNGCs.

post-infection, bMφ tended to form multinucleated giant cells (MNGCs) especially when infected with Mbv (Fig 2A). A quantitative analysis of MNGC numbers confirmed that Mbv infection induced more MNGCs when compared to Mtb-infected or non-infected (NI) cells (Fig 2B). Multinucleation of bMφ occurred only with live bacteria since paraformaldehyde-killed-Mbv (PFA-*Mbv*) showed similar numbers of MNGCs to both Mtb-infected and uninfected Mφ (S3A and S3B Fig). This effect was restricted to bMφ, since MNGCs were almost absent in hMφ infected either with Mtb or Mbv (Fig 2C). Importantly, Mbv-infected bMφ showed a broad range of nuclei associated with an increase of cell size (S3C Fig). In contrast, the limited number of multinucleated hMφ detected were

mostly binucleated with a small variation of cell size (S3C Fig). At the ultrastructural level, we confirmed that the multiple nuclei were contained in a single cell (Figs 2D and S3D). MNGCs were significantly larger than mononucleated bMϕ and contained a central vesicle-rich area surrounded by intact nuclei (Fig 2D). Although the number of nuclei was variable, their circular alignment at the periphery of the cytosol was reminiscent of the morphology of the multinucleated giant cells (also called Langhans' cells) present in granulomatous lesions (Figs 2D and S3D). Altogether, these data suggest that as early as 24 h post infection, Mbv interactions with bMϕ induces the formation of MNGCs. Importantly, 54 and 58% of MNGCs observed by fluorescence microscopy were uninfected, suggesting a bystander effect (Fig 2E and 2F). To test this possibility, we stimulated naïve bMϕ with filtered supernatant from infected-bMϕ. Only supernatants from Mbv-infected bMϕ led to a significant increase in MNGC formation when compared to supernatants from non-infected or Mtb-infected bMϕ (S4A Fig). In agreement with a species-specific phenotype, supernatants from hMϕ infected with either Mtb or Mbv failed to induce multinucleation in bMϕ (S4A Fig). These results demonstrate that factors present in the extracellular medium contribute to multinucleation in bMϕ. The factor(s) responsible for cell multinucleation were proteinaceous since heat-inactivated supernatant did not induce MNGC formation (S4A Fig). Similar experiments in infected hMϕ showed that none of the conditions tested increased the number of MNGCs (S4B Fig). These data show that proteins secreted by bMϕ after infection with Mbv induced MNGC formation.

## Secreted bacterial MPB70 contributes to MNGC formation in bMϕ

One of the main differences between Mbv and Mtb is the level of expression of genes in the SigK regulon that encode secreted proteins such as MPB70. The gene encoding MPB70 is highly expressed by Mbv, whereas its expression is low in Mtb *in vitro* but shows intracellular induction [41]. Furthermore, MPB70 contains a FAS1-domain, a structure that is known to play a role in cell adhesion [42,43]. Hence, we hypothesised that MPB70 induces multinucleation during Mbv macrophage infections [11,14,44,45]. To investigate whether MPB70 affects bMϕ multinucleation, we generated an Mbv strain with the gene encoding MPB70 deleted (Mbv ΔMPB70) and the complemented strain (Mbv-Compl) (Figs 3A and 3B and S5). As expected, deletion of the MPB70 gene completely abolished the secretion of MPB70 in Mbv ΔMPB70 compared to Mbv wild type (Mbv WT) and Mbv-Compl (Fig 3A). Next, we infected bMϕ with Mbv WT, Mbv ΔMPB70 or Mbv-Compl and analysed the numbers of MNGCs after 24 h of infection (Fig 3B). We found that the deletion of MPB70 markedly impaired the ability of Mbv to induce multinucleation. In agreement with MPB70 having a role in the multinucleation process, infection with Mbv-Compl induced MNGC formation at similar levels to that of Mbv WT infection (Fig 3C). We concluded that MNGC formation in bMϕ is a specific response to MPB70 activity after Mbv infection.

MPT70 (homologous to MPB70 in human-adapted strains) is expressed at a very low level by Mtb strains [11,14]. We then examined whether MPB70/MPT70 itself controled MNGC formation or whether other factors were required for bMϕ multinucleation. For this, we generated an Mtb strain constitutively expressing MPT70 (Mtb-MPT70+; S5E and S5F Fig). Next, we infected bMϕ with Mbv WT, Mtb WT or Mtb-MPT70+ for 24 h, and analysed the numbers of MNGCs (Fig 3D and 3E). As observed before, Mbv WT induced MNGCs and the number of MNGCs was significantly higher in bMϕ. Interestingly, the number of MNGCs in bMϕ infected Mtb-MPT70+ remained similar to that counted in both uninfected or Mtb WT infected cells (Fig 3D and 3E). This suggests that, unlike Mtb, Mbv induces a specific bMϕ response, which together with MPB70 expression enables the formation of MNGCs.

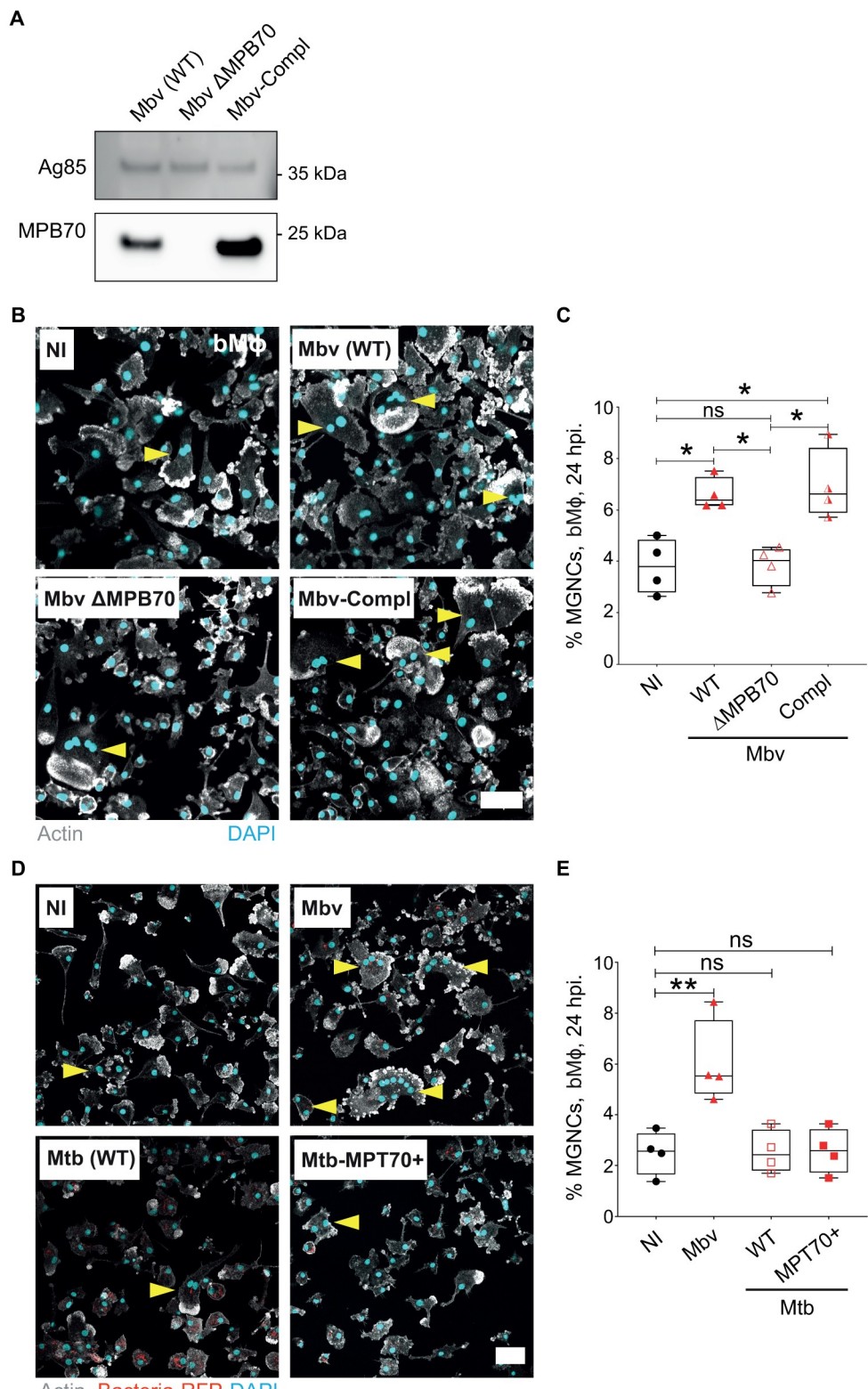

**Fig 3. Secreted bacterial MPB70 contributes to MNGC formation in bMφ.** (A) Culture filtrate from Mbv wild-type (WT), Mbv ΔMPB70, Mbv ΔMPB70/MPB70 (Mbv-Compl.) were assessed for immuno-labelling of MPB70 (23 kDa). Immuno-labelling of Ag85 (38 kDa) was used as gel loading control (B) Fluorescence confocal images of bMφ infected

with Mbv WT, Mbv ΔMPB70, Mbv-Compl. for 24 h. Uninfected cells had been used as a negative control. Cell nuclei are stained with DAPI (cyan) and actin is visualized in grey. Yellow arrows point the multi-nucleated cells. Scale bar, 50 μm (C) Quantification of MNGCs from each experiment displayed in B. Graph represents the quantification of MNGCs for each condition tested where each dot represents one bovine donor. Data shown are representative of four biological repeats. (D) Fluorescence confocal images of bMφ infected with Mbv WT, Mtb, Mtb-MPB70+ for 24 h. Uninfected cells had been used as a negative control. Cell nuclei were stained with DAPI (cyan); bacteria-RFP and actin are visualized in red and grey, respectively. Yellow arrows point the multi-nucleated cells. Scale bar, 50 μm (E) Quantification of MNGCs from each experiment displayed in D. Graph represents the quantification of MNGCs for each condition tested where each dot represents one bovine donor. Data shown are representative of two independent experiments.

## Extracellular vesicles from Mbv-infected bMφ promote MNGC formation

To identify the host factors involved in bMφ multinucleation, we analysed the secretome of Mtb- versus Mbv-infected bMφ. We identified 1341 host proteins in the cell-free supernatants and found that 192 proteins showed differential abundance between mycobacteria-infected bMφ and uninfected control samples by mass spectrometry-based proteomics (Fig 4A and S1 and S2 Tables). Notably, the majority of the identified proteins were associated with extracellular vesicle (EVs) biogenesis and trafficking, cellular focal adhesion or membrane trafficking (Fig 4A). Furthermore, 27 proteins were found to be differentially secreted between Mtb- and Mbv-infected bMφ (Fig 4B and S1 and S2 Tables). Given the marked presence of an EV-associated signature in the proteomic analysis, we hypothesized that EVs released during infection might contribute to bMφ multinucleation. To test this, we isolated EVs and assessed the quality and purity of the EV-enriched fractions by electron microscopy (Fig 4C and 4D). Naïve bMφ were then stimulated for 24 h with the EV-enriched fraction from uninfected (NI), Mtb- or Mbv- infected bMφ, or with PBS as a negative control. The percentage of MNGCs significantly increased in bMφ stimulated with the EV fraction from Mbv-infected bMφ, whereas the EVs from Mtb-infected bMφ did not induce significant MNGC formation (Fig 4E and 4F). Altogether, our data suggest that Mbv-induced EVs contribute to bMφ multinucleation by acting as carrier for the transport and delivery, to targeted cells, of key components involved in the multinucleation process.

## Granulomas from Mbv-infected cattle contain a higher number of MNGCs than Mtb-infected cattle

Inflammation occurring within granulomatous lesions leads to the formation of specific multi-nucleated cells, called Langhans' cells. Our *in vitro* data suggested that cattle infected with Mbv or Mtb should show variation in Langhans' cells *in vivo*. Post-mortem histopathological analysis was performed on H&E stained thoracic lymph node (LN) sections from cattle infected with Mtb H37Rv for 10 weeks or with Mbv AF2122/97 for 6 weeks. Limited numbers of granulomas were detected in LNs from cows infected with Mtb (Fig 5). In agreement with previously described studies [40,46], Mtb-induced granulomas were at an early stage of maturation (type I and II as described by [47,48]), with limited traces of necrosis or caseification (Fig 5A and 5C). In contrast, LNs from cows infected with Mbv contained a large number of granulomatous lesions at all stages of maturation (Fig 5B and 5C).

In agreement with the *in vitro* data, we found that the number of MNGCs present in Mtb-induced granulomas was significantly lower than in Mbv-induced granulomas (Fig 5A, 5B and 5E). It is noteworthy that for a similar amount of Acid Fast-positive Bacilli (AFB) detected in stage II granulomas from both Mtb- and Mbv-infected cows (Fig 5D), Mbv-infected granulomas contained around 2.5 times more MNGCs compared to Mtb-infected granulomas (Fig 5E). Moreover, the number of MNGCs increased during granuloma maturation, suggesting

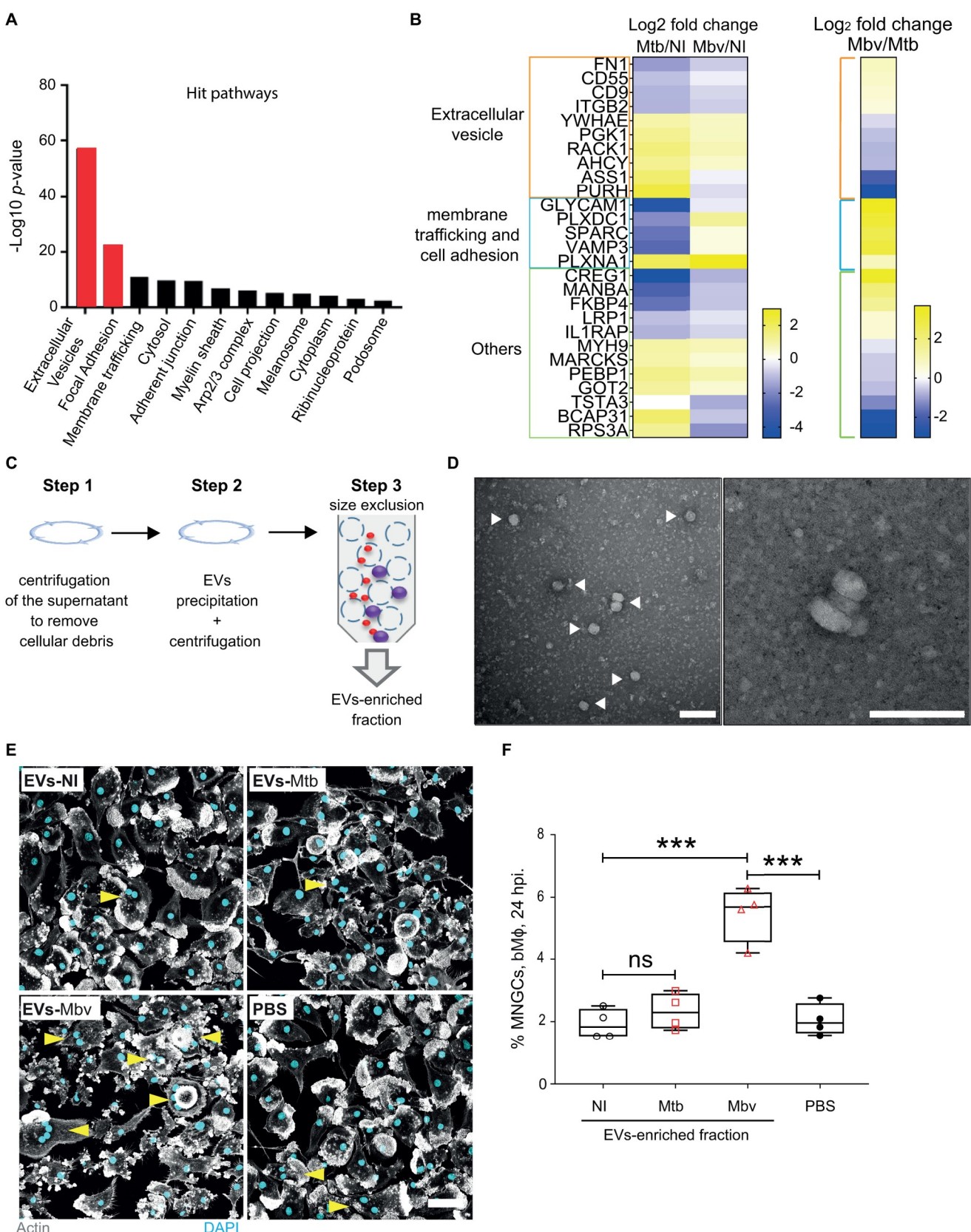

**Fig 4. Extracellular vesicles produced by Mbv-infected bMϕ induce cell multinucleation.** (A) A total of 192 proteins were determined to be differentially regulated using an ANOVA and Tukey's post-hoc test in the secretome of infected bMϕ and sorted according to Gene Ontology Cellular Component (GOCC). (B) List of 27 proteins differentially expressed in the secretome of Mtb-infected or Mbv-infected bMϕ clustered into 3 groups: extracellular vesicles, membrane trafficking and cell adhesion or others. The $\log_2$ fold change Mtb or Mbv, both normalised to uninfected (NI), samples are displayed in the left panel and the $\log_2$ fold change Mbv/Mtb is displayed in the right panel. Proteomics data were obtained from 3 independent experiments carried out with 3 different bovine donors. (C) EV purification procedure summarized in 3 steps. (D) Electron microscopy micrographs of EV-enriched fraction. The white arrows show the extracellular vesicles. Scale bar: 0.1 μm. (E) Fluorescence confocal images of naïve bMϕ stimulated for 24 h with EV-enriched fraction from uninfected (NI), Mtb-infected or Mbv-infected bMϕ. As EVs were purified in PBS, a similar volume of PBS was added to the bMϕ as a control. Cell nuclei are stained with DAPI (cyan) and actin is visualized in grey. Yellow arrows point the MNGCs. Scale bar: 50 μm (F) Quantification of MNGCs after bMϕ stimulation with EV-enriched fractions. The graph represents the quantification of MNGCs for each condition tested where each dot represents one bovine donor. Data are representative of 4 independent experiments.

that the expansion of the MNGC population correlates with the severity of granulomatous inflammation.

## Matured MNGCs provide a restrictive environment for *M. bovis*

In our model of bovine Mϕ, the MNGC population is heterogeneous, with cells showing different sizes and numbers of nuclei. A previous study reported that the differentiation process modified the anti-mycobacterial phenotype of MNGCs [29]. To define the effect of bovine Mϕ multinucleation on Mbv, we differentially evaluated the intracellular trafficking of Mbv in mononucleated cells, early MNGCs (containing 2–3 nuclei) or mature MNGCs (containing over 3 nuclei). First, we analysed Mbv intracellular burden in these three predefined cellular populations. Mononucleated and bi or tri-nucleated giant cells (defined as early MNGC) harboured a similar proportion of intracellular single bacteria as well as large bacterial clusters (Fig 6A). In contrast, MNGCs containing more than three nuclei (defined as mature) showed a significantly lower bacterial burden, mostly composed of single or small bacterial groups (Fig 6A). This suggests that mature MNGCs are either less permissive to Mbv and/or provide a more restrictive environment, limiting bacterial spread. In order to evaluate the nature of the intracellular environment encountered by Mbv in MNGCs, we analysed the intracellular localisation of Mbv in the three different macrophage populations. Interestingly, both mononucleated bMϕ and early MNGCs showed a similar proportion of Mbv positive for the lysosomotropic dye Lysotracker (LTR), indicating that the process of lysosomal acidification is maintained during the first stage of MNGC formation. Conversely, in mature MNGC, a significantly higher proportion of Mbv were localised in LTR positive compartments (Fig 6B). We next evaluated the proteolytic activity in the three defined Mϕ populations using the pan cysteine-cathepsin activity-based probe BMV109. Although the global cathepsin activity was high, we did not observe significant differences between the populations (Fig 6C). Because autophagy is a major anti-bacterial pathway, we finally monitored LC3B association with intracellular Mbv. As observed for acidification, LC3B association with Mbv was similar in both mononucleated bMϕ and early MNGCs but increased in mature MNGCs (Fig 6D).

We concluded that the major lysosomal degradative pathways remain functional during MNGC differentiation. Additionally, the increase of Mbv in acidified compartments and association with LC3B in mature MNGCs suggest that MNGCs present a more restrictive milieu for mycobacterial proliferation.

## Discussion

In this work, by analysing infections of Mϕ from two host species with two host-adapted mycobacteria, we shed new light on species-specific host-pathogen interactions in tuberculosis. When comparing the Mϕ infections side-by-side, unexpectedly, there were no striking

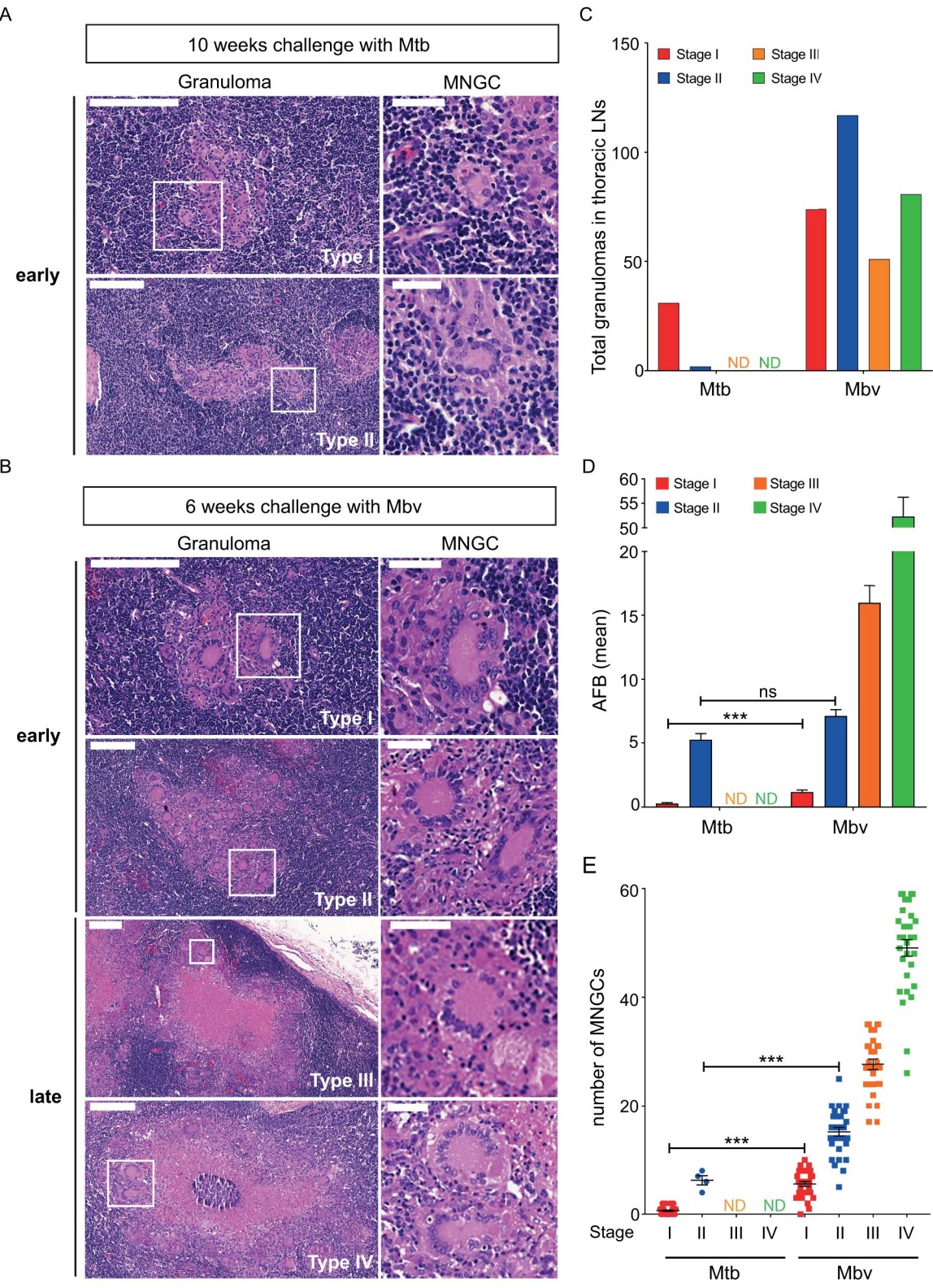

**Fig 5. Granulomas from Mbv-infected cattle contain a higher number of MNGCs than Mtb-infected cattle.** (A, left image) Histological Haematoxylin and Eosin (H&E) staining of granulomatous lesions in thoracic lymph nodes of cattle challenged for 10 weeks with Mtb H37Rv strains. Scale bar, 200 μm. (A, right image) higher magnification of multinucleated giant cells (white square). Scale bar, 50 μm. (B, left image) H&E staining of granulomatous lesions in thoracic lymph nodes of cattle challenged for 6 weeks with Mbv AF2122/97. Scale bar, 200 μm. (B, right image) higher magnification of multinucleated giant cells (white square). Scale bar, 50 μm. (A and B) Granulomas are classified from early stage of maturation (type I and II) to late granulomas (type III and IV) (C) Quantification of the number of granulomas counted for each condition. (D) Quantification of bacterial load in each granuloma based on acid fast staining of the tissue section and expressed as the mean Acid Fast Bacilli (AFB) ± SEM. (E) Quantification of the number of MNGCs per granuloma ± SEM.

species-specific differences in bacterial replication that could explain host tropism in human and bovine TB. Mtb replicates more efficiently than Mbv in hMφ, while both pathogens replicated similarly in bMφ. These differences could be related to the differentiation protocols and more studies are needed to identify the possible differentiation factors implicated. Nevertheless, both Mtb and Mbv showed similar uptake in each Mφ species, suggesting that despite the interspecies differences, both Mtb and Mbv are similarly recognized by macrophage receptors.

Mtb produces MNGCs in human disease and animal models, but the underlying mechanisms remain unclear. Here we show that both *in vitro* and *in vivo* the specific interaction between Mbv and the bovine host led to the formation of MNGCs, arguing that both host and pathogen factors are required for MNGC formation. Because the production and secretion of the mycobacterial protein MPB70 is one of the main phenotypic differences between Mbv and Mtb [14], this protein was an obvious candidate. Moreover, MPB70 has homology with the cell adhesion domain Fasciclin 1 (FAS1), Osteoblast-specific factor II (OSFII) or βIgH3, suggesting that MPB70 could mediate cell-to-cell contact or cellular adhesion by interacting with cellular receptors such as stabilins, integrins, as well as extracellular matrix components [42,43]. Cell-adhesion is a crucial step in the fusion process, bringing Mφ into close contact with extracellular matrix components. It is tempting to speculate that MPB70 is implicated in bMφ fusion by mediating cellular interactions and adhesion properties. The differential expression of MPB70 between both pathogens may also partially explain the inability of Mtb to induce MNGCs in our *in vitro* model. On this latter point, further studies are needed to explore the kinetics of MNGC formation *in vitro* and if, or how, Mtb can be modified to induce bMφ MNGC formation. Although MPB70 expression is low in Mtb, its expression may increase in stress conditions such as exposure to antibiotics or depending on the Mφ lineage in which the pathogen resides [49,50]. The MPB70 homologous protein MPT70 is thus likely secreted by *M. tuberculosis* and locally accumulates during human TB progression, contributing to the formation of MNGCs in human TB. The observed reduced levels of MNGCs in granulomas from cattle infected with Mtb as compared to Mbv further indicates that multinucleation requires both host and pathogen factors.

Macrophage fusion is a rare event that occurs in distinct environments [51]. The first study of MNGC formation in culture showed that human monocytes could differentiate into MNGCs when exposed to certain microenvironments [52]. Since then, numerous cytokines had been identified as potent inducers of MNGCs [21,23–25,53]. In most experimental models using monocytes, MNGC formation occurred after several days of exposure to fusogenic factors such as IL-4, suggesting that monocytes need to mature in order to acquire fusogenic properties [25,54]. In contrast, certain populations of primary Mφ alternatively activated by IL-4 can induce MNGC formation relatively rapidly (i.e. in hours) when exposed to additional pro-fusogenic stimuli [21]. Thus, the fusogenic properties of Mφ rely on multiple components, which include a specific inflammatory phenotype together with contact with other fusion partners. As MNGCs mostly originate from inflammatory Mφ [18], we used a model of GM-CSF-derived Mφ. We showed here that GM-CSF bMφ are competent to generate MNGCs specifically in response to Mbv infection. However, neither Mtb nor Mbv trigger MNGC formation

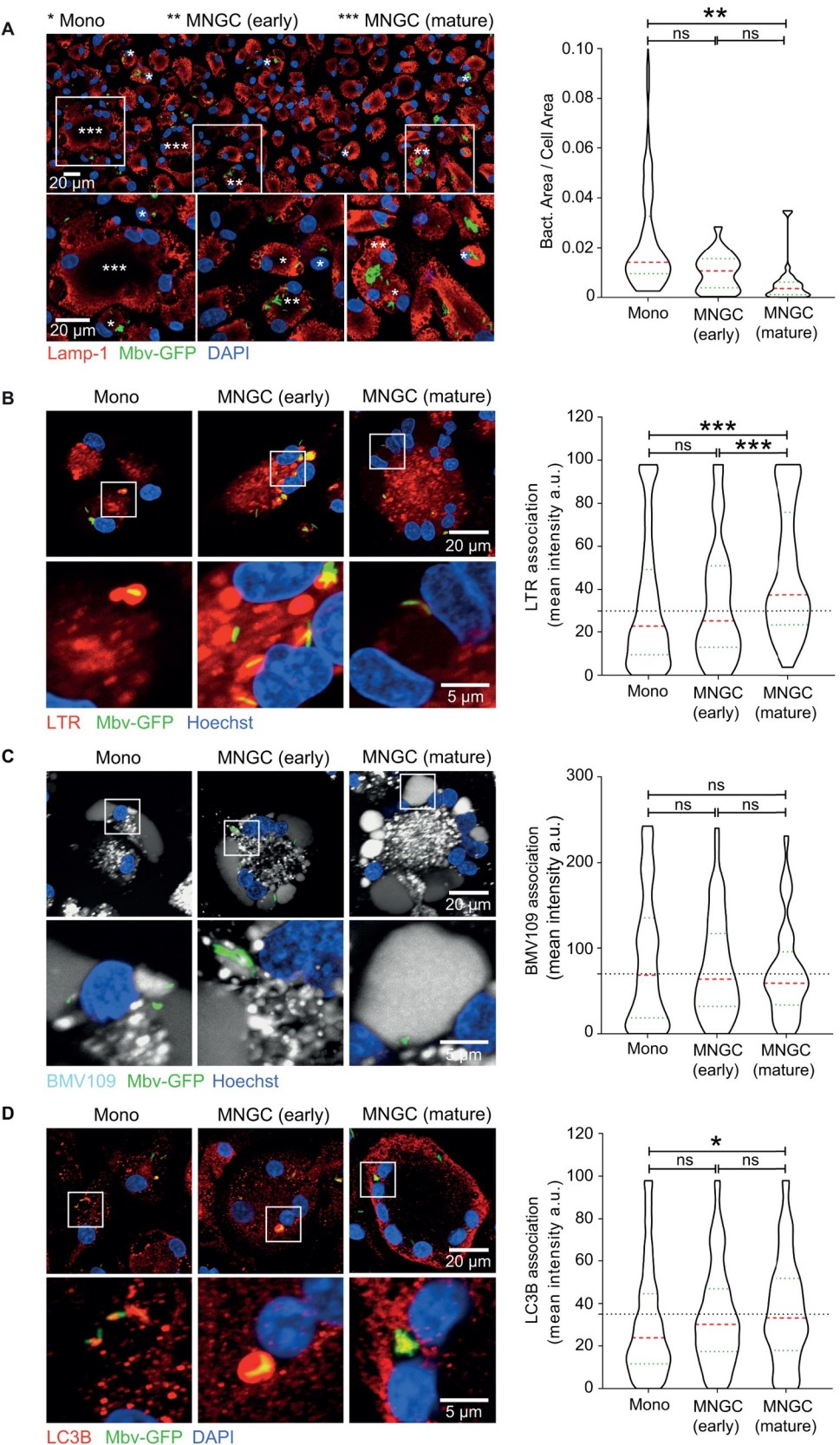

LTR Mbv-GFP Hoechst

BMV109 Mbv-GFP Hoechst

LC3B Mbv-GFP DAPI

**Fig 6. MNGCs provide a restrictive environment for Mbv.** (A) (upper panel) Tile scan of 3 successive confocal images showing the diversity of bMφ infected with Mbv-GFP for 24 h. Cells were fixed and immunolabelled for LAMP-1 (red), Mbv-GFP (green) and cell nuclei in blue (DAPI). The lower panel represents a crop of the region marked by a white square. The different cell population are marked as follow: *Mono: Mononucleated cells; **MNGC (early): Multinucleated cells with 2 or 3 nuclei; ***MNGC (mature): Multinucleated cells with over than 3 nuclei. Right panel: the quantification of the intracellular bacterial load in mononucleated and early or mature multinucleated cells is expressed in bacteria area per cell area. (Mono, n = 60 cells; MNGC (early), n = 22 cells; MNGC (mature), n = 28 cells). (B, C, D) bMφ infected with Mbv-GFP (displayed in green) for 24 h were stained with (B) LysoTracker Red (LTR, in red), (C) the pan cysteine-cathepsin activity-based probe BMV109 (in grey) and (D) immuno-labelled with a LC3B antibody (in red). DAPI was used to label nuclei (in blue). (B) LTR (number of bacterial object analysed (n)); Mono, n = 633; MNGC (early), n = 80; MNGC (mature), n = 88. (C) BMV109; Mono, n = 368; MNGC (early), n = 52; MNGC (mature), n = 76. (D) LC3B; Mono, n = 1374; MNGC (early), n = 243; MNGC (mature), n = 133. Each micrograph is a representative image of mononucleated cells (Mono), multinucleated giant cells (MNGC) with 2 or 3 nuclei (early) or over 3 nuclei (mature). The lower panels correspond to a zoom of the region marked by a white square. The quantification of each marker's association with bacteria is shown on the right panels. The dashed lines show the fluorescence cut-off applied. The data are representative from two independent experiments from 2–3 donors.

in hMφ. We speculate that in the context of human granuloma, MNGCs possibly originate from another subpopulation of activated Mφ recruited during TB inflammation.

Our results that GM-CSF-derived bMφ multinucleation is driven by factors released into the extracellular milieu after Mbv infection suggested a host factor was also involved. In this context, the extracellular microenvironment plays an important role in Mφ fusion and MNGC formation. The cell density, the nature of the extracellular matrix, and the local concentration of growth factors, chemokines or cytokines are important factors for cell differentiation, cell-to-cell contact and membrane fusion [55–57]. In agreement with this notion, the Mbv-infected bMφ secretome analysis revealed a dominant signature of proteins related to cell-adhesion, membrane trafficking, and extracellular vesicles. Extracellular vesicles and exosomes play a crucial role in cell communication, inflammation, and are implicated in multiple infectious diseases [58]. During Mtb infection, extracellular vesicles participate in the inflammation process through the recruitment of Mφ to the site of infection, the modulation of immune responses, antigen presentation, and/or transport of bacterial components [59–62]. Exosomes from murine J774 macrophages infected with Mtb contained several antigenic bacterial proteins such as EsxA, EsxB, SapM, PknG as well as MPT32, MPT53, MPT63 and MPT64 [59]. We speculate that EVs modulate cell multinucleation by acting as carriers for the transport of host and bacterial factors that will promote multinucleation. In our proteomics approach, we were not able to detect Mtb proteins and complementary approaches with increased sensitivity are needed for the identification of bacterial factors present in the supernatants. *In vivo*, when released, these vesicles may then traffic towards the tissue to another site such as the periphery of granulomas, interact with resident or recruited Mφ, and promote the formation of MNGCs.

During TB inflammation, Mφ fusion occurs during granuloma progression leading to MNGCs or Langhans' cells [17,63]. Generally, Langhans' cells are localised at the peripheral epithelioid rim of the granuloma and can contain around 20 nuclei. Langhans cells, associated with granulomatous lesions, are derived from pro-inflammatory macrophages (CD68+, CD40+, DC-STAMP+), which fuse in response to inflammatory stimuli such as macrophage-colony stimulating factor (M-CSF), Tumour necrosis factor-α (TNF-α) or interferon-γ (IFN-γ) [18,27,64,65]. Although the role of MNGCs in the granuloma is still unclear, several studies suggest a role of these cells in inflammation and bacterial control (reviewed in [18,66]). MNGCs can indeed take up large targets for degradation [67] and when they mature and lose their ability to uptake bacteria, the remaining NADH oxidase activity and antigen presentation clear all pathogens already inside [29]. Consistently, we found that mature MNGCs contained very low numbers of mycobacteria that were localised in acidic compartments with autophagic

features. Therefore, it is likely that during formation, MNGC evolve toward a more restrictive environment for intracellular mycobacteria.

Mtb has previously been isolated from lesions found in naturally infected cattle [68–71]. These latter cases show that, in certain circumstances, Mtb can cause disease in cattle, an outcome that may depend on the genetic background of both the infecting strain and host, as well as host immune status. This underlines the need to unravel the mechanistic basis for MNGC formation in well controlled experimental systems. In a head-to-head experimental cattle infection, we have previously shown that two Mtb strains, the hallmark Mtb H37Rv and an Mtb isolated from a bull in Ethiopia (Mtb BTB1558) were attenuated in cattle as compared to Mbv AF2122/97, with the latter inducing greater inflammation and pathology [11,40]. Here we found that, even when the amounts of bacteria are comparable, Mbv induces a higher number of MNGCs per granuloma than Mtb, including the early-stage type I and type II granulomas. This suggests that the intensity of the inflammatory response *in vivo* influences not only granuloma maturation but also the formation of MNGCs. Reduction of inflammation by treatment with anti-inflammatory IL-10 has been shown to reduce both granulomatous inflammation and MNGC formation [72]. Additionally, assessment of cytokine expression of MNGCs in cattle infected with Mbv showed a correlation between the expression of pro-inflammatory cytokines and the severity of granulomatous lesions [73]. The higher presence of MNGCs in Mbv thus correlates with its virulent phenotype in cattle, suggesting a role for MNGCs in inflammation, granuloma maturation and severity of TB lesions.

Altogether, our data provide evidence that a combination of bacterial effectors and species-specific responses shapes the singular interaction of the pathogen with its host, in this instance triggering the formation of MNGC, one of the cellular hallmarks of granulomas.

## Methods

### Ethics statement

Holstein Friesian cows housed at the RVC Boltons Park Farm (Hertfordshire, UK) were used to obtain blood for PBMC isolation and subsequent Mϕ generation. Animals were held at the RVC under certificate of designation. Animals were dewormed regularly, checked regularly for the presence of BVD and IBR, which can both infect bMϕ, leading to immunosuppression. Furthermore, all animals were negative for bovine TB using BOVIGAM IFN Gamma Test, which was conducted in addition to the standard tuberculin skin test. All study cows received a physical examination, including assessment of the respiratory, cardiovascular, gastrointestinal, musculoskeletal, and reproductive systems, as well as skin, udder, and teats. The physical examination included BCS, pulse rate, respiration rate, and a rectal temperature conducted by a licensed veterinarian or trained designee. To avoid influences on cellular function through various stages in the reproductive cycle, animals used were age- and lactation-matched (2nd or 3rd lactation, respectively), and were in mid-lactation. All procedures were carried under Home Office Project licence (PPL7009059), after approval by the RVC's Ethics and Welfare Committee. Furthermore, handling of cows and blood sampling were conducted in accordance with EU legislation (Directive 2010/63/UE, related to the protection of animals used for scientific goals).

### Reagents and antibodies

Recombinant human Granulocyte-Macrophage Colony Stimulating Factor (GM-CSF) and bovine GM-CSF were provided by Miltenyi (#130-093-867) and Abcam (#ab209168), respectively. DAPI was used to stain cell nuclei and was purchased from Sigma-Aldrich (#D9542) and the Alexa Fluor-488 Phalloidin was from Invitrogen (#A12379). Detection of cell death

was visualized using Live/Dead Green dye (ImmunoChemistry #6342). Immunolabelling of LC3B and LAMP-1 was performed using a rabbit anti-LC3B from Enzo Lifesciences (MBL-PM036) and a rabbit anti-LAMP-1 from Abcam (Ab24170). Mouse antibody to MPB70 (#LAB-0007a2) and the Mouse antibody to Ag85 (#ab36731) were purchased from Lionex GmbH and Abcam, respectively. The secondary antibodies used in this study were: goat anti-rabbit conjugated with Alexa Fluor 488 (#A11034), (ii) goat anti-mouse conjugated with Alexa Fluor 488 (#A11029), both purchased from Life technologies and (iii) an anti-mouse conjugated with HRP (#W402B) from Promega. LysoTracker Red DND-99 was from Thermofisher scientific (#L7528) and the cysteine cathepsin activity was determined using BMV109 (iABP Pan cathepsin (Cathepsin B/L/S/X)) from Vergent Bioscience (#40200). For light or electron microscopy, cells were fixed using Paraformaldehyde (PFA) from Electron Microscopy Sciences (#15710) or Glutaraldehyde from Sigma-Aldrich (#G5882), respectively.

## Mycobacterial strains

In the study, we used the sequenced and annotated *M. tuberculosis* H37Rv (Mtb) and *M. bovis* AF2122/97 (Mbv) reference strains as models [9,10,30]. Fluorescent Mtb and Mbv were engineered to constitutively express Red fluorescent proteins (RFP) encoded by the plasmid pML2570 integrated into the bacterial genome (Integrase: Giles, Resistance: Hygromycin). Mbv wild type (WT), Mbv *mpb70* knock-out *(*Mbv ΔMPB70) and the Mbv ΔMPB70 complemented strains (Mbv-compl) were used in this study. Mbv ΔMPB70 cloning and complementation are detailed in the paragraph "Cloning and characterisation of Mbv 2122/97 *mpb70* knock-out and complemented strains". Mtb strains were grown in 7H9 Middlebrook broth supplemented with 10% Albumin Dextrose Catalase (ADC), 0.05% Tween 80 and 0.5% glycerol. All Mbv strains were grown in 7H9 Middlebrook supplemented with 10% ADC, 0.05% Tween 80 and 40 mM sodium pyruvate. When required, selective antibiotics were added to the medium (S3 Table).

## Cloning and characterisation of Mbv 2122/97 *mpb70* knock-out and complemented strains

**Cloning strategy.** A knockout mutant in the *mpb70* gene (*Mb2900*) of Mbv 2122/97 was constructed using the phAE159 shuttle phasmid. One kb regions flanking the chromosomal Mbv *mpb70* were cloned into phAE159 so as to flank a hygromycin resistance marker to generate the allelic exchange substrate (S5A Fig and S3 Table). After amplification in *M. smegmatis* mc$^2$155, recombinant phages were used to infect Mbv wild type and hygromycin resistant Mbv ΔMPB70 transductants were selected. In the deletion mutant complementation was achieved by expression of the wild type *mpb70* gene from the replicating plasmid pEW70c2 (Mbv ΔMBP70::MBP70). In Mtb-H37Rv, *mpt70* from H37Rv was inserted in the plasmid pGM221-1. MPT70 constitutive expression is under the control of the *hsp60* promoter from the mycobacterial replicative plasmid pGM221-1 (S5E Fig and S3 Table).

**Selection and characterisation by PCR.** For PCR analysis, the bacteria were grown in 7H9 media containing 50 μg/ml hygromycin for Mbv ΔMPB70 cultures, and 50 μg/ml hygromycin and 25 μg/ml kanamycin for Mbv-Compl cultures. DNA for PCR analysis was collected through crude DNA extraction. PCRs were performed with the Phusion High-Fidelity DNA polymerase from NEB, using the Phusion GC Buffer. The primers listed below were used to ascertain the presence of *dipZ* and/or *mpb70* genes (S5A–S5C Fig) List of primers: *dipZ*-Forw: GAATTACCACGCCAAAGACG; *dipZ*-Rev: TCATCCGTAGGTGAAGGAAAA; *dipZ*-*mpb70*-intergenic-Forw: GCTCCGAAGAAATCATGTCG; *mpb70*-5'end-Rev: AGACAGCC ACCGCCAGAG; *mpb70*-Rev: CTGCGACATTCCCTGCAC.

**Detection of secreted MPB70 by Western blot.** The bacteria were grown in Sauton's media with antibiotics added as appropriate (described above). The supernatant of each strain was collected and concentrated with Amicon Ultra-15 Centrifugal Filter Units. Protein concentrations were then determined using a Pierce BCA Protein Assay kit and the supernatants were diluted to get a normalised whole protein load per well for gel electrophoresis of 25 μg. After transfer of the proteins to the membrane, the membrane was probed using a mouse anti-MPB70 IgG, and an anti-mouse HRP-conjugated (S5D and S5F Fig).

## Isolation of primary human and bovine monocytes and differentiation into macrophages

Human monocytes were obtained from leukocyte enriched blood fractions from healthy adult donors provided by the British National Health Service (NHS) under strict anonymity. Bovine blood was obtained from clinically healthy Holstein cows in mid-lactation phase (with a somatic cell count below 100,000 as an indicator of healthy udders) housed at Bolton Park's Farm of the Royal Veterinary College (University of London). All animals were regularly tested for freedom of TB infection (as per governmental guidelines) and were in addition free of Bovine Viral Diarrhoea Virus (BVDV) and Infectious Bovine Rhinotracheitis (IBR), both viral infections leading to immunosuppression. Blood was collected by jugular venepuncture into sterile glass bottles containing 10% Acid Citrate Dextrose (ACD) as anticoagulant, as previously described [74]. Prior to monocyte isolation, bovine whole blood was aliquoted into 50 mL centrifugation tubes and centrifuged for 15 min at 1200 x *g* to collect the buffy coat. Both human leucocyte enriched fraction and bovine buffy coat were then processed in the standard way but with the difference that PBS-EDTA was used as the buffer for the isolation of human monocytes, while PBS was used for the processing of bovine blood. Peripheral Blood Monocytes Cells (PBMCs) were isolated by centrifugation onto a Ficoll gradient. PBMCs were then washed and CD14-expressing monocytes were labelled with the anti-human CD14-antibody conjugated with magnetic beads (Miltenyi #130-050-201) and isolated by direct magnetic selection using the LS column (Miltenyi #130-042-401). Human and bovine CD14-positive monocytes were finally incubated in RPMI 1640 medium (Gibco # 72400021) containing: GlutaMax, 25 mM Hepes, 10% Foetal bovine serum (FBS) and 10 ng/mL of recombinant human GM-CSF or 20 ng/mL of recombinant bovine GM-CSF to allow differentiation into Mϕ for 7 days at 37°C in an atmosphere containing 5% $CO_2$.

## Cell culture

After 7 days of differentiation, when Mϕ were ready for infection and during all the infection process, GM-CSF was omitted from the culture medium. Once differentiated, MΦ were washed once with PBS and detached from the petri dish by incubation for 20 min at 4°C in PBS-EDTA followed by gentle scrapping. Cells were seeded depending on the type of experiment. For immunofluorescence, $1.8\times10^5$ cells/well were seeded into 24 well plates containing untreated glass coverslips (diameter 10 mm, No:1.5). For stereology, electron microscopy, transfer supernatant assays or RT-qPCR, cells were seeded into 6 well plates at a concentration of $7\times10^5$ cells/well. Alternatively, untreated glass bottom (No:1.5) dishes with grids from Matek Corporation (#P35G-1.5-14-C-GRID) containing $2\times10^5$ cells/dish were used for light and electron microscopy.

## Mycobacteria preparation and macrophage infection

Bacterial strains were cultured in a 50 mL centrifugal Falcon tube containing their respective optimal media until the exponential phase was reached. One day before infection, bacteria

were diluted to reach an Optical Density ($OD_{600nm}$) of 0.4 and cultured for an additional 24 h. Prior to infection, bacteria were pelleted by centrifugation at 2,900 x g for 5 min and washed twice with PBS. Bacteria were pelleted again before adding an equivalent number of sterile 2.5- to 3.5-mm glass beads that matched the pellet size into the Falcon tube (usually 4–5 beads). The Falcon tubes were then vigorously shaken for 1 min to break up bacterial clumps. The bacteria were suspended in 7 ml of RPMI 1640 medium containing GlutaMax, 25 mM HEPES, 10% FBS (complete RPMI) and transferred to a 15 mL centrifugal Falcon tube. Bacterial suspensions were then spun down at low speed (150 x g) for 3 min to allow the removal of the remaining bacterial clumps. The supernatant, cleared of bacterial clumps, was transferred into a clean 15 mL Falcon tube and the $OD_{600nm}$ was measured to determine the concentration of the bacterial suspension, and further diluted in complete RPMI to reach a final $OD_{600nm}$ of 0.1 (based on growth curves it was determined that an OD 0.1 the density of bacteria in the culture is $1 \times 10^7$ bacteria/ml). Bacteria were finally added to the MΦs at an MOI of 10 for hMϕ infection or an MOI of 1 for bMϕ infection and incubated at 37˚C in an atmosphere containing 5% $CO_2$ for 2h (bacterial uptake). Cells were then washed twice with PBS, the medium was replaced with fresh complete RPMI and incubated at 37˚C in an atmosphere containing 5% $CO_2$ until the samples were processed for analysis.

## Supernatant stimulation assay

Human or bovine Mϕ were seeded into 6 well or 24 well plates and infected as described above. The supernatants were then collected, sterilized by double filtration using 0.22 μm PVDF filters, before being transferred onto naïve uninfected Mϕ. Supernatant-stimulated Mϕ were then incubated for 24 h at 37˚C in an atmosphere containing 5% $CO_2$. For each experiment, infected Mϕ and supernatant-stimulated Mϕ were from the same donor. Finally, Mϕ were fixed overnight with a solution of 4% PFA- PBS before being stained for immunofluorescence.

## Extracellular vesicle (EV) purification

Two and a half million of bMϕ were seeded into T25 $cm^2$ flasks and incubated in complete RPMI, overnight at 37˚C in an atmosphere containing 5% $CO_2$. The following day, cells were infected with Mtb-RFP or Mbv-RFP at an MOI of 2. After 2 hours uptake, cells were washed 3 times with PBS to remove bacteria and residual FBS. Cells were then fed with RPMI 1640 without FBS (5 mL per T25 $cm^2$ flask) and incubated 24 hours at 37˚C in an atmosphere containing 5% $CO_2$. After 24 h infection, the supernatant from two T25 $cm^2$ flasks (total volume of at least 10 mL) was collected and processed for EV purification using the Exo-spin purification Kit from Cell guidance systems (#EX01). Briefly, the supernatants were cleared by ultracentrifugation at 16,000 x g for 30 min. Ten mL of cleared supernatants were then mixed with 5 mL of Exo-spin buffer and incubated overnight at 4˚C. EVs were then precipitated by centrifugation at 16,000 x g for 90 min. EVs pellets were then suspended in PBS and purified using a size exclusion chromatography resin column.

A small fraction of EVs were fixed with 1% Glutaraldehyde and 4% PFA-PBS solution and purity control was assessed by Electron microscopy. The rest of EV-enriched fractions were finally diluted in complete RPMI and used to stimulate naïve bMϕ. After 24 h of stimulation with EVs, cells were fixed with 4% PFA-PBS solution and the number of MNGCs was assessed by confocal light microscopy.

## Immunofluorescence and image acquisition

**Immunofluorescence.** Cells were cultured onto glass coverslips. Prior to immunolabeling, cells were fixed overnight at 4˚C with a 4% PFA-PBS solution. Fixed cells were then incubated

for 10 min at room temperature (RT) with a solution of PBS-NH$_4$Cl 50 mM to quench free alde-hyde groups, before being permeabilized for 20 min with PBS-Saponin 0.2%. Cells were then blocked for 30 min with PBS-BSA 1%. For the immunofluorescence, both primary and second-ary antibodies were diluted in PBS-Saponin 0.02%-BSA 0.1% (dilution buffer). LC3B and LAMP-1 were labelled using a rabbit anti-LC3 (dilution 1/200) and a mouse anti-LAMP-1 (dilu-tion 1/100). After 1 h incubation at RT, cells were washed three times with PBS and incubated with the respective secondary antibodies for one additional hour. Cells were washed 3 times prior to nuclei staining with DAPI. When required, the actin cytoskeleton was labelled using phalloidin conjugated with Alexa-Fluor 488 (dilution 1/800 into dilution buffer; incubation 45 min at RT). For the detection of live versus dead cells, the cells were stained with Green Live/Dead stain (500 nM for 10 min at RT) prior to PFA fixation. The coverslips were finally mounted onto microscopy glass slides using Dako fluorescence mounting medium (Dako, #S3023).

**Image acquisition.** Confocal images were acquired using a confocal inverted microscope (Zeiss LSM710) or Zeiss LSM880 both equipped with a 40X oil Lens or Plan-Apochromat 63x/1.4 NA lens and excitation laser 405, 488, 561, 633 nm. For the quantification of multinucle-ated cells, a field was randomly chosen, and images were acquired in tile scan mode (4x4) with 40X oil lens.

## Image analysis

Image analysis was performed using free open-source FIJI software (NIH). Zeiss LSM confocal images files (.czi) were opened using the BioFormats plug-in of FIJI.

**Cell number and intracellular bacterial growth.** Confocal images were first split into separate channels: Cell nuclei DAPI (blue), RFP (red) corresponding to RFP-mycobacteria.

(a) To measure the number of cells, a threshold was applied to the DAPI images in order to mask and calculate the number of nuclei. The presence of multinucleated cells in the field was manually adjusted for each image. (b) The bacterial area was calculated by applying the thresh-old function in the RFP-channel to mask the fluorescent bacteria. "Analyse Particles" function of FIJI (size = 0.5–infinity, circularity = 0–1) was applied to calculate the area of each bacteria. For each time point, the intracellular growth was expressed in bacteria (RFP) area per cell.

**Intracellular markers association.** Confocal images were first split into separate channels: Cell nuclei DAPI (blue), RFP-mycobacteria (red) and Green channel corresponding to the marker tested (e.g., LC3B). The cell number and bacteria area were determined as described in (a) and (b). Mask corresponding to all the bacteria was created and each bacteria object was extended from 2 pixels using the function "Dilate" and converted into a "region of interest". Finally, the green channel corresponding to the cellular marker was subjected to pixel intensity measurement within the bacteria region of interest. Marker association was expressed in mean intensity of green pixels for each bacteria object. For each marker, the fluorescence background was measured from several fields and bacteria were considered positive when the mean intensity of the marker > background + 1 STD. The percentage of association was calculated as follows: number of positive bacteria x 100 / total number of bacteria. As the marker LAMP-1 was also associated with spacious phagosomes in bMϕ, the same method was used to determine the LAMP-1 positive bacterial compartments with a manual correction for spacious LAMP-1 posi-tive phagosomes. The data were expressed as % of mycobacteria in LAMP-1 positive vacuole. All values were analysed and plotted using Excel and GraphPad Prism.

## Electron microscopy

**Extracellular vesicle negative staining.** Ten μl of sample was incubated at RT on a 200 mesh formvar/carbon grid. Grids were then washed 5 x 1 min in 200 mM HEPES (Sigma-

Aldrich H0887) and transferred to 20 μl 1% Uranyl acetate, (UA) (Agar scientific AGR1260A) and incubated for 1 min at RT. Excess 1% UA was removed and the grids were left to dry before imaging. Images were acquired using a 120 kV Tecnai G2 Spirit BioTwin (FEI company) with an Orius CCD camera (Gatan Inc.)

**Stereology.** Transmission electron microscopy sample preparation for stereology: Cells were washed in PBS and then fixed in 2.5% GA in 200 mM HEPES pH7.4 for 30 min at RT, followed by overnight fixation at 4°C. After several washes in 200 mM HEPES buffer, samples were processed in a Pelco Biowave Pro (Ted Pella, USA) with the use of microwave energy and vacuum. Briefly, samples were fixed and stained using a reduced osmium, thiocarbohydrazide, osmium (ROTO)/en bloc lead aspartate protocol. Samples for stereological analysis were dehydrated using an ethanol series of 50, 75, 90 and 100% then lifted from the tissue culture plastic with propylene oxide, washed 4 times in dry acetone and transferred to 1.5 ml microcentrifuge tubes. Samples were infiltrated with a dilution series of 50, 75, 100% of Ultra Bed Low Viscosity Epoxy (EMS) resin to acetone mix and centrifuged at 600 x g between changes. Finally, samples were cured for a minimum of 48 h at 60°C before trimming and sectioning. Sectioning and imaging: ultrathin sections (~50nm) were cut with an EM UC7 Ultramicrotome (Leica Microsystems) using an oscillating ultrasonic 35° diamond Knife (DiaTOME) at a cutting speed of 0.6 mm/sec, a frequency set by automatic mode, and a voltage of 6.0 volts. Images were acquired using a 120 kV Tecnai G2 Spirit BioTwin (FEI company) with an Orius CCD camera (Gatan Inc.)

Stereological analysis of Mtb infected cells: At least 22 different infected cells per group were imaged at a magnification of 3,900 by systematic and random sampling. Cross points of the stereological test grid over bacteria were counted with regard to the subcellular localization of bacteria, which was determined from images taken at a minimum magnification of x16,000. The following criteria were employed for the assessment of subcellular membrane involvement: (a) Single surrounding membrane; bacteria were, at least partially, tightly lined by a phospholipid bilayer, representing the phagosomal membrane (b) cytosolic; bacteria were surrounded by ribosomes, representing the cytoplasm with no indication of the phagosomal membrane; (c) Multiple surrounding membranes; bacteria were enveloped by double or multiple membrane structures. Data are shown as the proportions of the total counts per sample group.

## Real-time polymerase chain reaction (RT-qPCR)

hMϕ and bMϕ were seeded into 6 well plates (8 x $10^5$ cells per well) and infected as described earlier. Cells were washed three times with PBS before being lysed in Trizol (1 mL per well). mRNAs were purified using Direct-zol RNA Miniprep kit from Zymo Research (#R2052), following the manufacturer's recommendations and reverse transcribed to cDNA with Quanti-Tect Reverse Transcription Kit (Qiagen). Quantitative real-time RT–PCR (qRT–PCR) was performed using 11.25 ng cDNA per well with 0.5 μl TaqMan Gene Expression Assay probe and 5 μl TaqMan Universal PCR Master Mix in a 10-μl reaction volume on an Applied Biosystems QuantStudio 7 Flex Real-Time PCR System. Each reaction was performed in triplicate. Data was analysed using ExpressionSuite for QuantStudio (Applied Biosystems). Fold change was determined in relative quantification units using GAPDH for normalization of RT-qPCR. TaqMan probes used were from Thermo Scientific: human *IFN B1*-FAM (Hs01077958-s1), human *GAPDH*-FAM (Hs02786621-g1), bovine *IFN B1*-FAM (Bt03279050-g1) and bovine *GAPDH*-FAM (Bt03210913-g1).

## Secretome analysis of mycobacterial-infected bovine macrophages

**Sample preparation.** BMϕ were seeded into 6 well plates and infected with Mtb or Mbv as described above. Uninfected cells were used as control. After 2 h of bacterial uptake, cells

were washed three times with PBS and incubated in RPMI 1640 supplemented with glutamine but without FBS and without Phenol Red for 22 h. The supernatants (2 mL) were then collected and sterilized by double filtration using 0.22 μm PVDF filters. Supernatants were then placed at -80°C prior to being analysed.

**Filter-aided sample preparation and trypsin digestion of protein samples.** Samples were processed using a modified version of the filter-aided sample preparation (FASP) [75]. Briefly, 800 μl of cell-free supernatant was denatured in 4 M urea in 50 mM triethylammonium bicarbonate buffer (TEAB) and loaded onto a 0.5 mL Amicon ultra 30 kDa cut-off spin filter (Millipore). Samples were centrifuged at 12,000 x g for 15 min. Filters were washed thrice/twice by addition of 400 μL UB buffer (8 M urea in 50 mM TEAB) followed by centrifugation (12,000 x g, 15 min). This process was repeated twice for a total of three UB washes. Samples were reduced with 5 mM tris (2-carboxyethyl) phosphine at room temperature for 1 h and subsequently alkylated with 10 mM iodoacetamide for 1 h in the dark. Filters were then washed consecutively thrice/twice by addition of 400 μL UB buffer and twice by addition of 400 uL of TEAB followed by centrifugation (12,000 x g, 15 min). This process was repeated twice for a total of three UB washes. Four hundred μL of TEAB 50 mM was added to each filter, and the samples were centrifuged (12,000 x g, 15 min). This was repeated twice for a total of three TEAB washes.

Proteins were on-filter for 24 h at 37°C using 200 ng trypsin in a humidified chamber. The filter unit was placed in a new collection tube after digestion, and the peptides were obtained in the flow-through by centrifugation (12,000 x g, 15 min). Peptides were eluted once more from the filter unit by the addition of 250 μL 50mM TEAB and further centrifugation (12,000 x g, 15 min). Trypsin digestion was stopped with the addition of trifluoroacetic acid (TFA) at a final concentration of 1%, and peptides were desalted using Macro C18 Spin Columns (Harvard Apparatus). Peptides were dried before storage at -20°C.

**Mass spectrometry.** Peptides were dissolved in 2% acetonitrile containing 0.1% TFA, and each sample was independently analysed on an Orbitrap Fusion Lumos Tribrid mass spectrometer (Thermo Fisher Scientific), connected to an UltiMate 3000 RSLCnano System (Thermo Fisher Scientific). Peptides were injected on an Acclaim PepMap 100 C18 LC trap column (100 μm ID × 20 mm, 3μm, 100Å) followed by separation on an EASY-Spray nanoLC C18 column (75 ID μm × 500 mm, 2μm, 100Å) at a flow rate of 300 nL/min. Solvent A was 0.1% formic acid, 3% dimethyl sulfoxide in water, and solvent B was 0.1% formic acid, 3% dimethyl sulfoxide, 20% water in acetonitrile. The gradient used for analysis was as follows: solvent B was maintained at 3% for 5 min, followed by an increase from 3 to 35% B in 99 min, 35–90% B in 0.5 min, maintained at 90% B for 5 min, followed by a decrease to 3% in 5 min and equilibration at 3% for 10 min. The Orbitrap Fusion Lumos was operated in positive ion data-dependent mode for Orbitrap-MS and Ion trap-MS2 data acquisition. Data were acquired using the Xcalibur software package. The precursor ion scan (full scan) was performed in the Orbitrap in the range of 400–1600 m/z with a nominal resolution of 120,000 at 200 m/z. Ion filtering for Ion trap-MS2 data acquisition was performed using the quadrupole with a transmission window of 1.6 m/z. The most intense ions above an intensity threshold of $5 \times 10^3$ were selected for high-energy collisional dissociation (HCD). An HCD normalized collision energy of 30% was applied to the most intense ions, and fragment ions were analysed in the Ion trap using Rapid scan rate. The number of Ion trap-MS2 events between full scans was determined on-the-fly to maintain a 3 sec fixed duty cycle. Dynamic exclusion of ions within a ±10 ppm m/z window was implemented using a 35 sec exclusion duration. An electrospray voltage of 2.0 kV and capillary temperature of 275°C, with no sheath and auxiliary gas flow, was used. The automatic gain control (AGC) settings were $4 \times 10^5$ ions with a maximum ion accumulation time of 50 ms for Orbitrap-MS, and $1 \times 10^4$ ions with a maximum ion

accumulation time of 45 ms for Ion trap-MS2 scans, respectively. Ions with <2+ or undetermined charge state were excluded from MS2 selection.

**Mass spectrometry data analysis.** All tandem mass spectra were analysed using MaxQuant 1.6.1.6 [76], and searched against the *Bos taurus* proteome database (containing 23,965 entries) downloaded from Uniprot on 01 December 2018. Peak list generation was performed within MaxQuant and searches were performed using default parameters and the built-in Andromeda search engine [77]. The enzyme specificity was set to consider fully tryptic peptides, and two missed cleavages were allowed. Oxidation of methionine, N-terminal acetylation and deamidation of asparagine and glutamine was allowed as variable modifications. Carbamidomethylation of cysteine was allowed as fixed modification. A protein and peptide false discovery rate (FDR) of less than 1% was employed in MaxQuant. Proteins were considered confidently identified when they contained at least one unique tryptic peptide. Proteins that contained similar peptides and that could not be differentiated based on tandem mass spectrometry analysis alone were grouped to satisfy the principles of parsimony. Reverse hits, contaminants and protein groups only identified by site were removed before downstream analysis. A label-free quantification strategy was employed using the MaxLFQ algorithm [78] within MaxQuant. Assigned LFQ values of protein groups containing ≥2 unique peptides were used for statistical analysis in Perseus 1.6.2.3 [79]. Data were $log_2$ transformed and filtered to contain at least two valid LFQ values in one group for comparison. Missing values were imputed using random numbers drawn from a normal distribution that simulates signals from low abundance proteins. An analysis of variance (ANOVA) as performed, and p-values were corrected for multiple hypothesis testing using the Benjamini-Hochberg FDR method. A Tukey's post-hoc test was performed to determine pairwise comparisons among means of the different groups. A total of 192 differentially regulated proteins were identified using an ANOVA and Tukey's post-hoc test. Among those hits, 27 were differentially regulated between H37Rv and *M. bovis* groups. (The reproducibility between the different biological replicates is displayed in S5G and S5H Fig).

## Cattle infection and histopathological analysis

**Cattle infection.** Lymph node tissue samples used in this study were collected from animals infected in the context of a prior study [40]. Time constraints on the large animal biosafety containment facility meant that the total experimental duration was limited to 16 weeks; hence cattle had been infected with *Mtb* H37Rv for 10 weeks or *Mbv* AF2122/97 for 6 weeks at the 'Platform for experimentation on infectious diseases' biocontainment level 3 suites of the Institut National de la Recherche Agronomique (INRA), Tours, France, as previously described [40,80]. Briefly, eight female Limousin x Simmenthal cattle of approximately six months of age were divided into two groups of four. Animals were sedated with xylazine (Rompun 2%, Bayer, France) according to the manufacturer's instructions (0.2 mL/100 kg, IV route) prior to the insertion of an endoscope through the nasal cavity into the trachea for delivery of the inoculum through a 1.8 mm internal diameter cannula (Veterinary Endoscopy Services, U.K.) above the bronchial opening to the cardiac lobe and the main bifurcation between left and right lobes. For each strain, an infective dose of $1 \times 10^4$ CFU was delivered endo-bronchially in 2 ml of 7H9 medium. Two ml of PBS were used to rinse any remains of the inoculum into the trachea and then cannula and endoscope were withdrawn. The canal through which the cannula was inserted into the endoscope was rinsed with 20 ml of PBS and the outside of the endoscope was wiped with sterilizing wipes (Medichem International, U.K.) prior to infection of the next animal. Inoculation with the two different pathogens occurred at different days; the endoscope was sterilised as recommended by the manufacturer between the two the

infections. Retrospective counting of the inocula revealed infection with $1.66 \times 10^4$ CFU Mtb H37Rv and $1.12 \times 10^4$ CFU Mbv AF2122/97.

**Histopathological analysis.** Animals inoculated with Mtb H37Rv or Mbv AF2122/97 were sacrificed 10 weeks or 6 weeks post-infection, respectively and subjected to post-mortem analysis. In this study, tissues evaluated for gross pathology included the following, lymph nodes: left and right parotid, lateral retropharyngeal, medial retropharyngeal, submandibular, caudal, cranial mediastinal and cranial tracheobronchial and pulmonary lymph nodes. The presence of gross pathological TB-like lesions was scored as previously described [40,81]. Tissue samples were preserved in 10% phosphate-buffered formalin for 7 days before being embedded in paraffin wax. Four-micron sections were cut and stained with haematoxylin and eosin (H&E) or Ziehl-Neelsen staining for examination by light microscopy (at x100 magnification) to assess the number, developmental stage and distribution of each granuloma (types I-IV) [47,48], the number of Langhans' cells as well as assessing the quantity and location of acid fast bacilli for each granuloma within the tissue section.

## Statistical analysis

Results were plotted as mean ± SEM or SD and statistical analyses were performed in Microsoft Excel 2010 (Microsoft) and GraphPad Prism 8 (GraphPad Software Inc.). 2-tailed Student's t-tests were used to compare 2 groups and 1-way ANOVA with Tukey's multiple comparisons was used to compare 3 or more groups. A p-value is considered significant when $< 0.05$ and indicate as follow: $^*$p<0.05; $^{**}$ p<0.01; $^{***}$ p<0.001; ns: not significant.

## Supporting information

**S1 Fig. Physiological and morphological differences between GM-CSF-derived human and bovine macrophages.** (A) hMφ were infected with Mtb-RFP or Mbv-RFP at an MOI of 10, whereas bMφ were infected at an MOI of 1. Intracellular bacteria were quantified based on intracellular RFP signal and expressed in bacteria area per infected cells. Each dot represents the average of 10 fields from 2 independent experiments (also with different donors). (B) Evolution of the number of bMφ during the course of infection (2 to 72 hours post-infection). (C) Evolution of the number of hMφ during the course of infection (2 to 120 hours post-infection). (D) Left panel: Confocal images of hMφ infected with Mtb-RFP or Mbv-RFP after 5 days of infection. Brightfield was used to visualize the cells. Bacteria are visualized in red, cell nuclei were stained with DAPI (blue) and nuclei from dead cells in green. Scale bar: 20 μm. Right panel: Quantification of the level of cytotoxicity based on Green Live/Dead stain; uninfected. Non-infected cells (NI) were used as a control, n represents the number of cells analysed. (E) Left panel: Confocal images of hMφ infected with Mtb-RFP or Mbv-RFP for 2 hours or 8 days. Actin (in green) was used to visualize the cells. Cell nuclei were stained with DAPI (blue) and bacteria-RFP are visualized in red. Scale bar: 20 μm. Right panel: quantification of intracellular growth expressed in bacteria area (μm$^2$) per hMφ. Data are representative of 2 independent experiments. (F and G) hMφ and bMφ after 7 days differentiation with GM-CSF. (F) Representative electron microscopy images of uninfected hMφ and bMφ. Scale bar: 5 μm (G) Representative confocal images of uninfected hMφ or bMφ fluorescently stained for the late endosomal marker Lamp-1. The regions in the white squares are highlighted on the right-hand side of the micrograph.
(TIF)

**S2 Fig. Differential intracellular trafficking between Mtb and Mbv.** hMφ or bMφ infected with Mtb-RFP or Mbv-RFP (A, B, E and F) or Mtb-GFP or Mbv-GFP (C and D) for 24 h.

Samples were fixed and fluorescently stained for the late endosomal marker Lamp-1 (A and B), for the pH sensitive dye LysoTracker DN99 Red (LTR) (C and D) and for the autophagic marker LC3B (E and F). For each fluorescent confocal image, the cell nuclei were stained with DAPI. Positive association of bacteria with the different markers, delimited by a white square, are magnified and displayed at the top right corner and the right-hand side of each image. Scale bars represent 10 μm. Graphs represent the quantification of the marker association with Mtb or Mbv ± SEM from three independent experiments. Each dot represents the mean relative fluorescent intensity of the cellular marker with a single or distinct bacteria group. The population within each dotted red box corresponds to the percentage (± STD) of bacteria positive for the marker tested.
(TIF)

**S3 Fig. Mbv induces multinucleation of bMϕ.** (A) Fluorescence confocal images of bMϕ infected with Mtb-RFP, Mbv-RFP or PFA-killed-Mbv-RFP for 24 h. Non-infected cells (NI) were used as a control. The bacteria are visualized in red, the cells-actin cytoskeleton is in white (phalloidin-488) and cell nuclei (DAPI) in cyan. The white square represents a region of interest magnified below each image. Scale bar, 40 μm. (B) Quantification of the percentage of MNGCs in bMϕ for each condition. Data are representative of two independent biological repeats, each carried out in duplicate. (C) GM-CSF-bMϕ or -hMϕ were infected with Mbv-RFP (red) for 24 h. PFA-fixed infected cells were stained with phalloidin-Alexa Fluor 488 (actin, green). DAPI (blue) was used to stain nuclei. Images presented above were acquired using a confocal microscope. Images were analysed using Harmony software (PerkinElmer). Actin stain was used to mask the cell bodies and determine the number and the area of the cells detected. DAPI staining was used to segment and count the number of nuclei in each cell. Cells containing 2 or more nuclei were considered. For each cell represented by a grey dot, values were plotted as nuclei number as a function of the cell's area (μm2). Scale bar, 50 μm (D) Electron microscopy image of Mbv-induced bovine MNGCs containing three or six distinct nuclei (red arrows). Scale bar, 10 μm.
(TIF)

**S4 Fig. Supernatant transfer assay for the determination of soluble factors involved in MNGCs formation in bMϕ.** (A) Supernatant transfer assay from bMϕ (non-infected (NI), infected with Mtb or Mbv), or from hMϕ infected with Mtb or Mbv for 24 h, onto naïve bMϕ. A fraction of supernatant from bMϕ-infected with Mbv was heat inactivated and used to stimulate naïve bMϕ (heat Mbv). Graph represents the number of MNGCs formed in cultures of naïve bMϕ following the addition of 400 μl of supernatant derived from cultures of bMϕ infected with Mbv or Mtb (B) Supernatant transfer assay from non-infected hMϕ (NI), hMϕ-infected with Mtb, Mbv, or bMϕ infected with Mtb or Mbv for 24 h, onto naïve hMϕ. A fraction of supernatant from hMϕ-infected with Mbv was heat inactivated and used to stimulate naïve hMϕ (heat Mbv). The graph represents the quantification of MNGCs for each condition tested. (A and B) Data are representative of two independent experiments, each carried out in duplicates.
(TIF)

**S5 Fig. Construction of Mbv MPB70 mutants and secretome analysis.** (A) Genetic arrangement Mbv wild-type (WT) and Mbv mpb70 knock-out (Δmpb70) strains. (B) Genetic organisation of *dipZ-mpb70* genes and primer locations (C) Characterisation of Mbv mutants by PCR. Deletion of mpb70 and maintenance of *dipZ* in Mbv Δmpb70 and complemented compared to WT. *dipZ* is amplified in all the strains. *mpb70* (244 bp) was amplified in the WT strain but not in Δmpb70 and complemented strains. The PCR with a forward primer in *dipZ*

and reverse primer in mpb70 gave the expected amplification of a 370 bp product for WT, but was absent in Δmpb70. Its absence from the complemented Δmpb70/mpb70 (Compl) strain, confirms that the location of the *mpb70* gene, carried by the replicative plasmid pEW70c2, is distal to the wild type chromosomal location. (D) Western immunoblot for detection of MPB70 in the supernatant of the Mbv WT, Δmpb70 and complemented (Mbv-Compl) strains. Each sample was analysed in duplicate. A 23 kDa band corresponding to MPB70 was detected for the WT and complemented strain but not for Δmpb70 strain. (E) Graphical map of the plasmid pGM221-1 used to generate the strain Mtb overexpressing MPT70 (Mtb-MPT70+). The Mtb-H37Rv gene *mpt70* is expressed under the control of hsp60 promotor. The clonality of the Mtb-MPT70+ strain was sustained by the Kanamycin resistance cassette carried by the plasmid pGM221-1. (F) Western immunoblot for detection of MPB70 and its homologous MPT70 in the supernatant of the Mbv WT, Δmpb70, Mbv-Compl, Mtb WT and Mtb-MPT70 + strains. A 23 kDa band corresponding to MPB70 was detected for Mbv WT, Mbv-Compl and MTB-MPT70+ strains but not for Mbv Δmpb70 and Mtb WT strains. (G) Principal component analysis of secretome samples. Distinct clusters of sample groups are observed that indicate high reproducibility between replicates and major protein expression differences between the samples analysed. (H) Unsupervised hierarchical clustering of secretome samples. Distinct clusters of sample groups indicate high reproducibility of replicates and distinct protein expression differences between the samples. Two major clusters are observed with uninfected controls clustering separately from mycobacterial-infected samples. Protein groups with lower abundance (e.g., lower LFQ intensity) are in purple and protein groups with higher abundance are in orange.
(TIF)

**S1 Table. Proteins secreted by bMϕ during infection with Mtb or Mbv.**
(XLSX)

**S2 Table. Mass spectrometric raw data of the secretome of non-infected (NI), Mtb- or Mbv-infected bMϕ.**
(XLSX)

**S3 Table. Mycobacterial constructs.**
(DOCX)

# Acknowledgments

We thank the Crick Science Technology platforms for support and advice during this work, Rocco D'Antuono and Kurt Anderson (Light Microscopy); Matt Russell and Lucy Collinson (Electron Microscopy). We thank Apoorva Bhatt for his critical advice and Chris Davies and Kat Pacey for technical assistance.

# Author Contributions

**Conceptualization:** Christophe J. Queval, Stephen V. Gordon, Maximiliano G. Gutierrez.

**Data curation:** Christophe J. Queval, Tiaan Heunis, Matthias Trost, Maximiliano G. Gutierrez.

**Formal analysis:** Christophe J. Queval, Antony Fearns, Waldo Garcia-Jimenez, Tiaan Heunis, Matthias Trost.

**Funding acquisition:** Stephen V. Gordon, Maximiliano G. Gutierrez.

**Investigation:** Christophe J. Queval, Maximiliano G. Gutierrez.

**Methodology:** Christophe J. Queval, Antony Fearns, Bernardo Villarreal-Ramos, Waldo Garcia-Jimenez, Tiaan Heunis, Matthias Trost, Dirk Werling, Francisco J. Salguero, Stephen V. Gordon, Maximiliano G. Gutierrez.

**Project administration:** Christophe J. Queval, Maximiliano G. Gutierrez.

**Resources:** Christophe J. Queval, Antony Fearns, Laure Botella, Alicia Smyth, Laura Schnettger, Morgane Mitermite, Esen Wooff, Bernardo Villarreal-Ramos, Waldo Garcia-Jimenez, Tiaan Heunis, Matthias Trost, Dirk Werling, Francisco J. Salguero, Stephen V. Gordon, Maximiliano G. Gutierrez.

**Software:** Christophe J. Queval, Antony Fearns, Waldo Garcia-Jimenez, Tiaan Heunis, Matthias Trost, Francisco J. Salguero.

**Supervision:** Christophe J. Queval, Maximiliano G. Gutierrez.

**Validation:** Christophe J. Queval, Tiaan Heunis, Matthias Trost, Francisco J. Salguero, Stephen V. Gordon, Maximiliano G. Gutierrez.

**Visualization:** Christophe J. Queval.

**Writing – original draft:** Christophe J. Queval, Maximiliano G. Gutierrez.

**Writing – review & editing:** Christophe J. Queval, Antony Fearns, Bernardo Villarreal-Ramos, Tiaan Heunis, Matthias Trost, Dirk Werling, Francisco J. Salguero, Stephen V. Gordon, Maximiliano G. Gutierrez.

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
