## [Decision Letter · Decision Letter 0]

14 Dec 2020

Dear Dr Gutierrez,

Thank you very much for submitting your manuscript "Macrophage-specific responses to human- and animal-adapted tubercle bacilli reveal pathogen and host factors driving multinucleated cell formation" for consideration at PLOS Pathogens. As with all papers reviewed by the journal, your manuscript was reviewed by members of the editorial board and by several independent reviewers. In light of the reviews (below this email), we would like to invite the resubmission of a significantly-revised version that takes into account the reviewers' comments.

We cannot make any decision about publication until we have seen the revised manuscript and your response to the reviewers' comments. Your revised manuscript is also likely to be sent to reviewers for further evaluation.

Sincerely,

Marcel A. Behr

Associate Editor

PLOS Pathogens

Sabine Ehrt

Section Editor

PLOS Pathogens

Kasturi Haldar

Editor-in-Chief

PLOS Pathogens

orcid.org/0000-0001-5065-158X

Michael Malim

Editor-in-Chief

PLOS Pathogens

orcid.org/0000-0002-7699-2064

Reviewer's Responses to Questions

**Part I - Summary**

Reviewer #1: The authors have made a considerable effort to respond to my comments (and even more so to those of the other reviewers!).

I would like to make it clear that my assessment of their work as "descriptive and preliminary" was certainly not intended to downplay the importance of the quantity and quality of their work.

Now, in their responses to my comments and those of the other reviewers, the authors have convinced me that this work represents a sufficiently interesting and important body of work for the community, although in my opinion it provides almost more questions than answers, but that is the beauty of science.

The work is solid, the data is solid, its interpretation could sometimes be debated, but at least this work will have the merit of nourishing reflection in this difficult and as yet little explored field.

Reviewer #2: M. tuberculosis and M.bovis, two closely related pathogens have interesting, poorly understood differences in host tropism. This study is aimed at understanding these differences using the each of the two pathogens in bovine and in human macrophages and assessing differences in infection phenotypes that include replication, subcellular localization, multinucleated giant cell (MNGC) formation. The problem with the work is the lack of a link of any of these findings to known differences in natural infection or to any new insights that might explain such differences. The authors have tried to link their in vitro findings of differences to the extent of MNGC formation by M. bovis versus M. tuberculosis in cows. However, even if these differences correlated with the ones they see in vitro, it is not clear how this is relevant to natural infections. After all, as the authors point out, M. tuberculosis causes MNGC formation in humans as does M. bovis in cows. In a second example of lack of link or insight from their findings, the authors show that M. tuberculosis replicates better than M. bovis in human macrophages and both replicate similarly in bovine macrophages. This might be the opposite of what one might expect, given that M. bovis is the pathogen with the broad host range. In summary, while it is clear that a great deal of work has gone into these experiments, unfortunately these findings do not add to our understanding of this interesting area of mycobacterial pathogenesis and evolution.

Reviewer #3: The different in vivo outcome of Mtb and Mbovis has been known since the 19th century. Yet, the reason for this difference in virulence has remained elusive. In this paper, Queval and colleagues propose that Mbovis, but not Mtb, induces the formation of MNGCs, in a MPB70-dependent manner, but that this phenotype is only observed in bovine macrophages (and not human macrophages).

What makes this exciting is that it is a seminal observation that could enable the mechanistic study of virulence traits that are specific to MTBC variants. What makes this potentially confusing is that it is unclear whether it is a host- and pathogen- or a (host+pathogen)-driven phenotype. This latter question is critical for moving the field forward after this paper. However, because of the different levels of infection and the error bars, it was not entirely clear what phenotypes were specific to only a select group and what phenotypes might be more broadly observed. As examples, the only time one sees 5% MNGCs in Hmacs is with M. bovis. Many points are around zero, but can we be convinced that this only happens with Bmacs, or rather that the Bmacs are simply turned on more, such that all phenomena are more observable? The data in S4D suggests this is possible. Here, bMACs show an increase in MNGC from 4 (non infected group) to 8 with M. bovis (one star). In the same Figure, hMacs infected with M. bovis see the MNGC go from 4 (non infected) to 7 (ns); beside this, hMAcs infected with M.tb have a median of 4 MNGC (no stats shown comparing M. bovis to Mtb).

**Part II – Major Issues: Key Experiments Required for Acceptance**

Reviewer #1: (No Response)

Reviewer #2: (No Response)

Reviewer #3: I can envision two key experiments to help clarify this issue and resolve whether this phenotype truly depends on the host-pathogen mix and a third experiment that could potentially lock down the role of MPB70.

1) In 3F, if the phenotype requires bMac EVs, can the authors prepare bMac EVs in a pathogen-independent manner, e.g. after phagocytosis of latex beads, to show whether the host cell on its own can produce the MNGC phenotype?

2) In 3F, to test whether the M. bovis pathogen on its own can result in MNGC formation, can the authors simply prepare a culture filtrate and determine whether the culture filtrate, on its own, results in MBNGC formation?

3) As a modification on this second experiment, can the authors prepare a culture filtrate of M. bovis disrupted for MPB70 to determine whether a bacterial-induced MNGC phenotype is diminished when MPB70 is absent?

**Part III – Minor Issues: Editorial and Data Presentation Modifications**

Reviewer #1: (No Response)

Reviewer #2: (No Response)

Reviewer #3: I don’t understand why the Y-axis is jumping around. IN 2B, the BMacs go up to 15 but the HMacs go up to 20. Since the results are lower with the Hmacs, why is the axis scaled to a higher number? In 4F it goes to 8. In S3B it goes up to 40.

There are times when the bMac data is presented before the hMac, and times it goes the other way around. I propose it will be more readable if there is a standard convention. This also applies to the writing, where some sentences presented the data in reverse order for variety, but it can lead to confusion.

PLOS authors have the option to publish the peer review history of their article (what does this mean?). If published, this will include your full peer review and any attached files.

Reviewer #1: No

Reviewer #2: No

Reviewer #3: No
---

## [Editor Report · Decision Letter 1]

19 Feb 2021

Dear Dr Gutierrez,

We are pleased to inform you that your manuscript 'Macrophage-specific responses to human- and animal-adapted tubercle bacilli reveal pathogen and host factors driving multinucleated cell formation' has been provisionally accepted for publication in PLOS Pathogens.

Best regards,

Marcel A. Behr

Associate Editor

PLOS Pathogens

Sabine Ehrt

Section Editor

PLOS Pathogens

Kasturi Haldar

Editor-in-Chief

PLOS Pathogens

orcid.org/0000-0001-5065-158X

Michael Malim

Editor-in-Chief

PLOS Pathogens

orcid.org/0000-0002-7699-2064
---

## [Editor Report · Acceptance letter]

10 Mar 2021

Dear Dr Gutierrez,

We are delighted to inform you that your manuscript, "Macrophage-specific responses to human- and animal-adapted tubercle bacilli reveal pathogen and host factors driving multinucleated cell formation," has been formally accepted for publication in PLOS Pathogens.

Best regards,

Kasturi Haldar

Editor-in-Chief

PLOS Pathogens

orcid.org/0000-0001-5065-158X

Michael Malim

Editor-in-Chief

PLOS Pathogens

orcid.org/0000-0002-7699-2064